# On the Dynamic Regret of Following the Regularized Leader: Optimism with History Pruning

**Naram Mhaisen** [1]    **George Iosifidis** [1]

## Abstract

We revisit the Follow the Regularized Leader (FTRL) framework for Online Convex Optimization (OCO) over compact sets, focusing on achieving dynamic regret guarantees. Prior work has highlighted the framework's limitations in dynamic environments due to its tendency to produce "lazy" iterates. However, building on insights showing FTRL's ability to produce "agile" iterates, we show that it can indeed recover known dynamic regret bounds through optimistic composition of future costs and careful linearization of past costs, which can lead to pruning some of them. This new analysis of FTRL against dynamic comparators yields a principled way to interpolate between greedy and agile updates and offers several benefits, including refined control over regret terms, optimism without cyclic dependence, and the application of minimal recursive regularization akin to AdaFTRL. More broadly, we show that it is not the "lazy" projection style of FTRL that hinders (optimistic) dynamic regret, but the decoupling of the algorithm's state (linearized history) from its iterates, allowing the state to grow arbitrarily. Instead, pruning synchronizes these two when necessary.

## 1. Introduction

This paper addresses the Online Convex Optimization (OCO) problem (Zinkevich, 2003; Shalev-Shwartz, 2012; Hazan, 2022), a popular paradigm for sequential decision making under uncertainty. OCO is regarded as a time-slotted game between a learner and a potentially adversarial environment. At slot $t$, the learner selects an action $\boldsymbol{x}_t$ from a convex set $\mathcal{X}$. Then, the environment reveals a cost

$f_t : \mathcal{X} \subset \mathbb{R}^n \to \mathbb{R}$, and the learner incurs $f_t(\boldsymbol{x}_t)$. The objective is finding a set of actions $\{\boldsymbol{x}_t\}_{t=1}^T$, for some horizon $T$, that perform well under the functions $\{f_t(\cdot)\}_{t=1}^T$, which is non-trivial since $\boldsymbol{x}_t$ is committed before $f_t(\cdot)$ is revealed. Under the most stringent criteria, the learner's performance is measured by the *dynamic regret* $\mathcal{R}_T$:

$$\mathcal{R}_T \doteq \sum_{t=1}^T f_t(\boldsymbol{x}_t) - f_t(\boldsymbol{u}_t),$$

where $\{\boldsymbol{u}_t\}_{t=1}^T \in \mathcal{X}$ is any set of comparators with desirable costs that we wish to benchmark against. The dynamic regret is thus simply the performance gap between the learner and the comparator sequence. A key complexity measure associated with the comparators is its *path length* $P_T$,

$$P_T \doteq \sum_{t=1}^{T-1} \|\boldsymbol{u}_{t+1} - \boldsymbol{u}_t\|. \tag{1}$$

The path length quantifies the variation in the comparator sequence, with larger values indicating a more dynamic environment and a more challenging learning setting.

### 1.1. Background and Motivation

Algorithms with a sub-linear regret guarantee have been behind recent state-of-art advances not only in classical computer science problems such as caching (Mhaisen et al., 2022), portfolio management (Tsai et al., 2024), and generalized assignment (Aslan et al., 2024), but also in machine learning problems such as the design of (enhanced) ADAM optimizer (Ahn et al., 2024), sub-modular optimization (Si-Salem et al., 2024), and supervised learning with shifting labels (Bai et al., 2022) among others. However, even in the *static* settings where the comparator is fixed: $\boldsymbol{u}_t = \boldsymbol{u}, \forall t$, the strategy of minimizing witnessed costs at each $t$ admits *linear* regret, indicating a failure in learning (Shalev-Shwartz, 2012, Sec 2.2). Hence, careful *regularization* is needed to avoid overfitting past data, which is the fundamental idea behind the two main algorithmic families for OCO: Follow the Regularized Leader (FTRL) and Online Mirror Descent (OMD). It is known that both frameworks achieve an order optimal static regret bound of $\mathcal{O}(\sqrt{T})$ (Orabona, 2022, Sec. 5). Further, it has been shown that when the regularization is made *data-dependent*, the regret bounds will

---

[1]Faculty of Electrical Engineering, Mathematics and Computer Science, TU Delft, Netherlands. Correspondence to: Naram Mhaisen <n.mhaisen@tudelft.nl>.

*Proceedings of the 42$^{nd}$ International Conference on Machine Learning*, Vancouver, Canada. PMLR 267, 2025. Copyright 2025 by the author(s).

indeed be sub-linear and, more importantly, data-dependent. This is a desirable, albeit challenging, objective.

Data-dependent bounds are preferable because they are parametrized by the actual problem instance, $\{f_t(\cdot)\}_{t=1}^T$, rather than crude universal bounds on this instance. I.e., on the Lipschitz constant $L$ and the horizon $T$. While this complicates the analysis, it leads to more custom algorithms in which the "easiness" of an instance is reflected in the bound. For example, AdaGrad-style bounds (Duchi et al., 2011), and follow-ups, achieve static regret of the form $\mathcal{O}(\sqrt{G_T})$ where $G_T = \sum_{t=1}^T \|g_t\|^2$, $g_t \in \partial f_t(x_t)$ is the sub-gradients norm trajectory. This form of bounds is desirable since they scale with the lengths $\|g_t\|$ and hold for any value $T$ rather than being dependent on the worst case $L$ value and on a single pre-provided $T$. Yet, they maintain the order-optimal bound ($\mathcal{O}(\sqrt{T})$) in all cases. Similarly, (Chiang et al., 2012) achieves $\mathcal{O}(\sqrt{V_T})$ for smooth functions where $V_T \doteq \sum_{t=2}^T \max_x \|\nabla f_t(x) - \nabla f_{t-1}(x)\|^2$ is the gradient variation trajectory, which, again, is never more $\mathcal{O}(\sqrt{T})$ but tighter for slowly-varying functions.

A more general problem-dependent quantity is the accumulated *prediction error* (Rakhlin & Sridharan, 2013; Mohri & Yang, 2016); suppose the learner receives a prediction $\tilde{f}_t(\cdot)$ for the cost function $f_t(\cdot)$ *prior* to deciding $x_t$, with no guarantees on its accuracy, the quantity of interest is:

$$E_T \doteq \sum_{t=1}^T \epsilon_t^2, \quad \epsilon_t \doteq \|g_t - \tilde{g}_t\|, \tag{2}$$

where $\tilde{g}_t \in \partial \tilde{f}_t(x_t)$. Clearly, we can choose $\tilde{f}_t(\cdot)$ to be 0 or $f_{t-1}(\cdot)$, recovering the dependence on $G_T$ and $V_T$[1], respectively. Algorithms whose regret depends on $E_T$ are called *optimistic* and are crucial for achieving best-of-both-worlds-style guarantees: *constant* regret in predictable environments and sub-linear in all cases, delivering adaptability without sacrificing robustness. Interestingly, the application of optimistic algorithms extends beyond enabling the use of untrusted predictions in the OCO problem; they have been shown to be key in the more general *delayed* OCO problem (Flaspohler et al., 2021), as well as the related OCO with memory problem (Mhaisen & Iosifidis, 2024). Given their significance, we focus on developing "optimistic" algorithms in this work. That is, algorithms that receive and use predictions of future costs and have regret bounds parametrized by $E_T$.

While (optimistic) data dependence is well understood for both OMD and FTRL frameworks under the *static* regret metric, the story is different when it comes to dynamic regret. For OMD, the current prevalent form of optimistic problem-dependent dynamic regret bounds first

---

[1] More precisely, for differentiable functions, we recover $V_T' = \sum_{t=1}^T \|\nabla f_t(x_t) - \nabla f_{t-1}(x_{t-1})\|^2 \leq V_T$.

appeared in (Jadbabaie et al., 2015), who used a variant of the Optimistic OMD (OOMD) (two-step variant). This formulation requires that $g_t$ is defined before calculating $x_t$, which is only possible for linear functions (recall $g_t \in \partial f_t(x_t)$). This "cyclic dependency" issue was only addressed recently in (Scroccaro et al., 2023), who obtained bounds that depend on a quantity similar to $E_T$, $D_T \doteq \sum_{t=1}^T \|\nabla f_t(y_{t-1}) - \nabla \tilde{f}_t(y_{t-1})\|$, where $y_t$ are points generated by an online algorithm.

As for FTRL, the first dynamic regret guarantee has been established by (Ahn et al., 2024), showing $\mathcal{R}_T = \mathcal{O}(P^{1/3}T^{2/3})$ for bounded domains. While this guarantee is not data-dependent and suboptimal in $T$, it suffices for the authors' goal of explaining the behavior of the Adam optimizer. For bounded domains, to our knowledge, no prior work has established dynamic regret guarantees for FTRL, problem-dependent or not, with $\mathcal{O}(P_T^\beta \sqrt{T})$ dependence, for any $\beta \in [0, 1]$. This gap raises an intriguing question regarding the performance of FTRL under the dynamic regret metric, particularly given that FTRL can be *equivalent* to OMD under specific regularization and linearization choices (McMahan, 2017, Sec. 6), suggesting its potential applicability in dynamic environments. However, the extent to which FTRL admits meaningful dynamic regret guarantees remains an open problem.

Conceptually, the versatility of FTRL arises from its *richer* state representation, where the "state" refers to the vector used to determine the next iterate, $x_{t+1}$. In OMD, the state of the algorithm is merely the current feasible point, $x_t$. In contrast, the state in FTRL is some mapping of all previous cost functions. In the most common case, this mapping is simply the cumulative gradient, $g_{1:t} \doteq \sum_{\tau=1}^t g_\tau$. This same versatility, which stems from retaining all past costs, introduces a key drawback: retaining all past costs can hinder adaptation when the costs are nonstationary.

Specifically, it has been demonstrated that FTRL iterates that use such mapping *fail* to achieve sublinear dynamic regret even for *constant* path lengths (Jacobsen & Cutkosky, 2022, Thm. 2). In essence, the aggregation of past costs obscures the switching patterns in the data (i.e., in $\{f_\tau(\cdot)\}_{\tau=1}^t$), which are crucial for the iterates to adapt appropriately, particularly when competing with moving comparators. Since most FTRL variants in the literature aggregate past gradients, FTRL is often considered unsuitable for dynamic environments (Chen et al., 2024). However, these findings do not dismiss the potential of *all* FTRL variants. On the contrary, they give insight into how to design variants that adapt to changing comparators, a key motivation behind the pruning mechanism we analyze here.

This paper seeks to address the ambiguity regarding FTRL's performance under dynamic regret. Clarifying this issue is not only intellectually compelling but also important for

establishing more refined problem-dependent bounds, as we demonstrate in the sequel. It will also help explain the notable performance gap between lazy (i.e., typically FTRL) and greedy (i.e., typically OMD) methods in dynamic settings. Furthermore, we introduce a new set of FTRL-native analysis tools, expanding its applicability in dynamic environments and paving the way for dynamic regret guarantees in other OCO settings (e.g., delayed feedback or memory constraints), where FTRL is often the framework of choice.

### 1.2. Methodology and Contributions

As noted earlier, certain forms of FTRL are equivalent to OMD, suggesting that dynamic regret guarantees should hold in these cases. The equivalence arises when the update minimizes the regularized linearized history, which is the starting point of this paper. Specifically, let $r_t(\cdot)$ be a data-dependent strongly convex regularizer that evolves with $t$, potentially depending on past costs and actions. Then, the standard linearized FTRL update is

$$\boldsymbol{x}_{t+1} = \operatorname*{argmin}_{\boldsymbol{x} \in \mathcal{X}} \langle \boldsymbol{g}_{1:t}, \boldsymbol{x} \rangle + r_{1:t}(\boldsymbol{x}). \qquad (3)$$

If $\mathcal{X} = \mathbb{R}^n$, and $r_t(\boldsymbol{x})$ is proximal[2], this update is equivalent to the OMD update which uses $r_{1:t}(\cdot)$ as the mirror map. This is proven in (McMahan, 2017, Thm. 11), even for the more general case of composite costs.

Here, we consider the *optimistic* version of this update. That is, we append the prediction $\tilde{f}_{t+1}(\boldsymbol{x})$ to the sum in (3). This will later allow us to have problem-dependent bounds that are modulated by $E_T$. Secondly, we focus on compact sets $\mathcal{X} \subset \mathbb{R}^n$. A primary design choice in our method is to incorporate the set constraint as an additional indicator function to each prediction. This is equivalent to modeling each cost function as a composite function: $f_t(\boldsymbol{x}) + I_{\mathcal{X}}(\boldsymbol{x})$. The indicator part is then always assumed to be "predicted" perfectly. Nonetheless, the linearization of the past *composite* costs is now different. Namely, our proposed update becomes:

$$\boldsymbol{x}_{t+1} = \operatorname*{argmin}_{\boldsymbol{x}} \langle \boldsymbol{p}_{1:t}, \boldsymbol{x} \rangle + r_{1:t}(\boldsymbol{x})$$
$$+ \tilde{f}_{t+1}(\boldsymbol{x}) + I_{\mathcal{X}}(\boldsymbol{x}), \quad (4)$$

with the state vector $\boldsymbol{p}_{1:t}$ calculated as the aggregation of

$$\boldsymbol{p}_t = \boldsymbol{g}_t + \boldsymbol{g}_t^I,$$
$$\boldsymbol{g}_t \in \partial f_t(\boldsymbol{x}_t), \quad \boldsymbol{g}_t^I \in \partial I_{\mathcal{X}}(\boldsymbol{x}_t) = \mathcal{N}_{\mathcal{X}}(\boldsymbol{x}_t),$$

where $\mathcal{N}_{\mathcal{X}}(\boldsymbol{x})$ is the normal cone at $\boldsymbol{x}$, and is defined as

$$\mathcal{N}_{\mathcal{X}}(\boldsymbol{x}) \doteq \{\boldsymbol{g} \mid \langle \boldsymbol{g}, \boldsymbol{y} - \boldsymbol{x} \rangle \leq 0, \forall \boldsymbol{y} \in \mathcal{X}\}.$$

----
[2]A proximal regularizer $r_t(\boldsymbol{x})$ is minimized at $\boldsymbol{x}_t$.

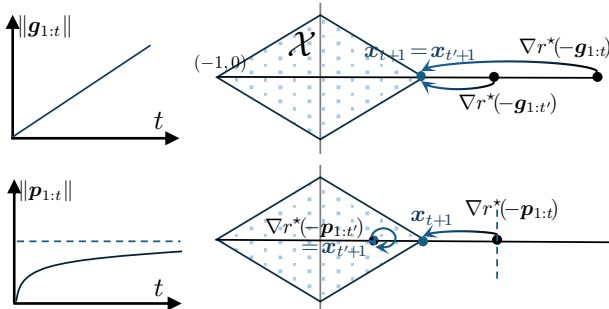

Figure 1: Effect of dual state size on iterates agility. We consider two slots $t < t'$, where gradients switch direction: $\boldsymbol{g}_\tau = (-1, 0)$ for $\tau \leq t$, and $\boldsymbol{g}_\tau = (1, 0)$ for $\tau > t$. **Top**: Standard FTRL accumulates a large state $\boldsymbol{g}_{1:t}$, and the update via $\nabla r^\star(\boldsymbol{g}_{1:t})$ becomes insensitive to the change in direction; both $\boldsymbol{g}_{1:t}$ and $\boldsymbol{g}_{1:t'}$ map to the same iterate. **Bottom**: A well-maintained state $\boldsymbol{p}_{1:t}$ remains bounded, and hence its mapping stays close to $\mathcal{X}$, enabling $\boldsymbol{x}_{t'}$ to start aligning quickly with the new better iterate direction $(-1, 0)$.

The simple yet key observation is that with the extra flexibility provided by $\boldsymbol{g}_t^I$, the state of the algorithm need not be the simple aggregation $\boldsymbol{g}_{1:t}$. Rather, some of the summands can be attenuated or *pruned* by carefully selecting $\boldsymbol{g}_t^I$ from the cone $\mathcal{N}_{\mathcal{X}}(\boldsymbol{x}_t)$, This is possible since $\mathcal{N}_{\mathcal{X}}(\boldsymbol{x}_t)$ contains all (scaled) negative subgradients of the expression in (4) when $\boldsymbol{x}_t$ lies on the boundary of $\mathcal{X}$ $\boldsymbol{x}_t \in \mathbf{bd}(\mathcal{X})$ (from the optimality conditions for constrained problems, see e.g., (Beck, 2017, Thm. 3.67)). In words, when an iterate leaves the feasible set and is thus projected back, we can choose to prune the state that led to this situation and replace it with an *alternative* state $\boldsymbol{p}_{1:t}$, that induces the same iterate but is smaller in norm. This construction is crucial, as we later show that the norm of the state is the key bottleneck for FTRL when competing with time-varying comparators. Fig. 1 illustrates how different state constructions behave upon a switch in cost direction; note that the FTRL update, with a fixed $r(\cdot)$ can be expressed as the projection of $\nabla r^\star(-\boldsymbol{g}_{1:t})$, where $r^\star(\cdot)$ is the conjugate of $r(\cdot)$ (see def. in Appendix A.4).

Our main contribution is formalizing this intuition to provide a dynamic regret analysis of Optimistic FTRL, leading to a new variant, *Optimistic Follow the Pruned Leader* (OptFPRL). This variant achieves *zero* dynamic regret when predictions are perfect. This is because the quality of predictions controls *all* regret terms, including $P_T$. To our knowledge, this is the first variant to explicitly have such full dependence on prediction accuracy without oracle tuning. In the general case, OptFPRL also maintains the minimax optimal rate of $\mathcal{O}(\sqrt{(1 + P_T)T})$ when $P_T$ is known. We also present a version that does not require prior knowledge of $P_T$, but assumes observability, while still maintaining dependence on prediction errors.

Next, since OptFPRL follows an FTRL-style analysis, it is

well-suited to incremental regularization in the manner of AdaFTRL (Orabona & Pál, 2018). In this scheme, we set the regularization *recursively*, incrementing it by, roughly, the regret by time $t$ had we followed $\boldsymbol{x}_t, \forall \tau \leq t$ (local regret). This is different from the more common approach, which sets this regularization as an *upper-bound* on this local regret. While the two are nearly equivalent in the worst case, the former allows for more refined bounds by applying only the minimal necessary regularization at each slot. Unlike prior optimistic dynamic regret algorithms, this style of regularization offers a more granular bound.

In summary, we investigate FTRL's dynamic regret over compact sets, and identify the unbounded growth of the state as the main bottleneck. By leveraging a simple and general mechanism, pruning past gradients, we gain a new degree of freedom that enables improved optimistic bounds.

## 2. Related Work

**Optimism and (dynamic) regret**. The pursuit of data dependence has been a central theme in online learning research since the introduction of AdaGrad (Duchi et al., 2011; McMahan & Streeter, 2010). This has eventually led to the development of Optimistic FTRL (e.g., (Mohri & Yang, 2016)). For a comprehensive survey of data-dependent online learning, we refer readers to (Joulani et al., 2020). However, these studies focus on static regret. Although the dynamic regret metric has been part of the OCO framework since its introduction, the first data-dependent dynamic regret bound only appeared in (Jadbabaie et al., 2015). The authors established a general bound, from which the result $\mathcal{O}(\sqrt{(P_T^* + 1)(E_T + 1)})$ can be derived, where $P_T^*$ is specific to the sequence of minimizers comparators $\{\boldsymbol{u}_t = \operatorname{argmin}_{\boldsymbol{x}} f_t(\boldsymbol{x})\}$, but in fact the bounds hold for any sequence whose path is *observable* online. This is done via a specialized doubling trick for non-monotone quantities. By removing the assumption of observable path lengths (i.e., considering all sequences simultaneously), dynamic regret bounds typically lose the sublinear dependence on $P_T$. For instance, in (Jadbabaie et al., 2015), the bound becomes $\mathcal{O}((P_T + 1)\sqrt{E_T + 1})$.

To address this, (Zhang et al., 2018) proposed a meta-learning framework that achieves $\mathcal{O}(\sqrt{T(1 + P_T)})$ for any sequence, which is shown to be minimax optimal. This framework was made data-dependent in (Zhao et al., 2020), who obtained $\mathcal{O}(\sqrt{(V_T + P_T + 1)(1 + P_T)})$ bound, among others. In this framework, multiple sub-learners are employed, each implementing Online Gradient Descent (OGD) with a different (doubling) learning rate, tuned for various ranges of $P_T$. These sub-learners are then "tracked" by a variant of the Hedge meta-algorithm. This two-layer approach was later unified within the framework of Optimistic OMD and made more efficient in terms of the number of

gradient queries (Zhao et al., 2024). Overall, knowing $P_T$ in advance, or assuming it is observable online, allows us to tune the regularization (or learning rate for OMD) with this knowledge, getting the better dependence $\sqrt{P_T}$. Without such assumptions, we can use the meta-learning framework, which learns online the best such regularization/learning rate under smoothness assumptions.

Setting aside the order of dependence on $P_T$ for a moment, we examine the quantity $E_T$. The aforementioned studies suffer the shortcoming hinted at in the introduction: to make $E_T$ small, we require knowledge of $\nabla f_t(\boldsymbol{x}_t)$ at *the start* of time slot $t$, which is generally not feasible[3] even if access to $\nabla f_t(\cdot)$ was provided (i.e., perfect prediction). This is because $\boldsymbol{x}_t$ remains unknown at the start of $t$ — it is the very point being determined. While the bounds are still optimistic, they are not informative for the predictor design. In the literature, this "cyclic" issue was only identified in (Scroccaro et al., 2023, Sec. 1.B). Thus, the authors introduce another related quantity $D_T$, defined earlier, and obtain $\mathcal{O}((1 + P_T)(1 + \sqrt{D_T}))$.[4] In the FTRL variants we propose, we do not have the cyclic issue because we do not require linearizing the predictions $\tilde{f}_t(\cdot)$.

The aforementioned studies are based on OMD, particularly on the "two-steps" variant of (Chiang et al., 2012), where the learner selects two points in each iteration, an intermediate one and the actual action. This distinction from our work is not merely technical. For instance, in the related "OCO with delay" setting, prior work has established guarantees for Optimistic FTRL and the one-step version of Optimistic OMD proposed in (Joulani et al., 2020, Sec. 7.2). However, similar guarantees are not yet known for the two-step variant of OMD (Flaspohler et al., 2021), and it remains unclear whether the same analysis can be extended to them.

Additionally, focusing again on optimism, a common limitation in the aforementioned studies is that the $P_T$ quantity[5] in the bounds is unaffected by the predictions. That is, the $P_T$ term may appear independently of any controllable quantity such as $E_T$, leading to bounds of $\mathcal{O}(P_T)$ even under perfect predictions (or zero gradient variation). While this may partly reflect limitations in the existing analyses rather than the algorithms themselves, the current bounds do not fully capture the interaction between prediction quality and path length. In contrast, OptFPRL analysis reveals that $P_T$, or more precisely, each $\|\boldsymbol{u}_{t+1} - \boldsymbol{u}_t\|$, is multiplied by the prediction error (e.g., $\epsilon_t$ or $\sqrt{E_t}$). Thus, the effect of $P_T$ can be attenuated when predictions are accurate.

---

[3]Unless all functions are originally linear. Note that linearization does not solve this issue, as it is performed *after* the learner has committed to its action.

[4]The authors also derive bounds based on "temporal variation", but we do not consider this quantity here.

[5]Or $\sqrt{P_T}$, assuming a known budget or an observable sequence with a doubling trick.

**OMD, FTRL, and (linearized) history.** The interplay between OMD and FTRL has received increasing attention in recent years. (Fang et al., 2022) studied a modified version of OMD under the static regret metric and showed its equivalence to Dual Averaging, a time-adaptive instance of FTRL, up to some terms in the normal cone. In this paper, we show that it is precisely these normal cone terms that become critical to achieving dynamic regret guarantees. For dynamic regret, (Jacobsen & Cutkosky, 2022) provides a comprehensive study via "centered" OMD, which incorporates FTRL-like centering properties. Their focus is primarily on unbounded domains, where such centering is essential. While their work integrates FTRL features into OMD, we take the opposite approach: extending native FTRL results to dynamic settings. This approach reveals failure modes of FTRL in non-stationary environments and offers principled solutions. Specifically, we investigate how modulating FTRL's state, in bounded domains, leads to regret bounds that are fully modulated by prediction accuracy.

In fact, our update is mostly related to the following form

$$\boldsymbol{x}_{t+1} = \underset{\boldsymbol{x}}{\operatorname{argmin}} \langle \boldsymbol{g}_{1:t} + \boldsymbol{g}_{1:t-1}^{\psi}, \boldsymbol{x} \rangle + r(\boldsymbol{x}) + I_{\mathcal{X}}(\boldsymbol{x}),$$

$$\boldsymbol{g}_t^{\psi} \in \partial I_{\mathcal{X}}(\boldsymbol{x}_{t+1}),$$

which appears in (McMahan, 2017) under the name "FTRL Greedy", and was studied under the *static* regret metric. Beyond explicitly modeling the sub-gradient selection from the cone, which allows controlling when and what to prune, we extend this formulation by incorporating: $(i)$ function predictions, $(ii)$ prediction-adaptive regularization, and $(iii)$ recursive regularization inspired by AdaFTRL. These modifications require different analysis tools, especially under the *dynamic* regret metric.

FTRL variants that reduce dependence on history have recently been proposed by (Zhang et al., 2024; Ahn et al., 2024) using *geometric discounting* of all past costs, which is specifically designed for the metric of "discounted" regret and its applications. Interestingly, however, Ahn et al. (2024) also observes that such manipulation of FTRL's state can endow it with certain (not necessarily optimal) dynamic regret guarantees. Though their method differs, the core insight aligns with ours: limiting FTRL's memory is essential for adapting to non-stationarity.

We also note the existence of other approaches for modeling "optimism" in OCO, other than seeking $E_T$ dependence, such as the Stochastically Extended Adversary (SEA) model (Sachs et al., 2022) that was studied in (Chen et al., 2024) for OMD with dynamic comparators. In addition, there exists the "correlated hints" interpretation of optimism (Bhaskara & Munagala, 2023), where the prediction quality is measured by their correlation with the actual cost. This later line of work assumes *strongly* convex domains and obtains a $\mathcal{O}((1 + P_T) \log^2(T) \sqrt{B})$ bound, where $B$ is the number

---

**Algorithm 1** Optimistic Follow the Pruned Leader (OptFPRL)

**Input**: Compact set $\mathcal{X}$, strategy for selecting $\sigma_t, \forall t$.
**Output**: $\{\boldsymbol{x}_t\}_{t=1}^T$.
 1: set $\boldsymbol{x}_1 = \arg\min_x \tilde{f}_1(\boldsymbol{x}) + I_{\mathcal{X}}(x)$
 2: **for** $t = 1, 2, \ldots, T$ **do**
 3:     Use action $\boldsymbol{x}_t$
 4:     *($f_t(\cdot)$ is revealed)*
 5:     Incur cost $f_t(\boldsymbol{x}_t)$ and compute $\boldsymbol{g}_t \in \partial f_t(\boldsymbol{x}_t)$
 6:     Compute $\tilde{\boldsymbol{g}}_t \in \partial \tilde{f}_t(\boldsymbol{x}_t)$ and the error $\epsilon_t = \|\boldsymbol{g}_t - \tilde{\boldsymbol{g}}_t\|$
 7:     Calculate the parameter $\sigma_t$ using $\epsilon_t$.
 8:     Set $\boldsymbol{g}_t^I$ according to (5).
 9:     Compute the (pruned) vector $\boldsymbol{p}_t = \boldsymbol{g}_t + \boldsymbol{g}_t^I$.
10:     Receive prediction $\tilde{f}_{t+1}(\cdot)$.
11:     Compute $\boldsymbol{x}_{t+1}^{\text{uc}}$ by solving (4) without $I_{\mathcal{X}}(\cdot)$.
12:     Set $\boldsymbol{x}_{t+1} = \Pi(\boldsymbol{x}_{t+1}^{\text{uc}})$.
13: **end for**

---

of slots without correlation. Lastly, an interesting reduction from dynamic to static settings is explored in (Jacobsen & Orabona, 2024), which revealed a fundamental tradeoff between gradient variability and the comparator sequence's complexity measures (e.g., the path length considered here).

## 3. OptFPRL

In this section, we present the proposed algorithm and characterize its dynamic regret. The routine of OptFPRL is described in Alg. 1. It takes as input the compact convex set $\mathcal{X}$, along with a strategy for determining the regularization parameters $\sigma_t$ based on information available up to and including slot $t$.

The initial action is based on the first prediction. Then, upon executing each $\boldsymbol{x}_t$ (line 3), the true cost $f_t(\boldsymbol{x})$ is revealed and the subgradient $\boldsymbol{g}_t$ is computable (line 5). In line 6, we evaluate the prediction error $\epsilon_t$, which is used in line 7 to update the regularization parameter $\sigma_t$ of the regularizer $r_t(\cdot)$ according to some pre-determined strategy.

We use scaled Euclidean regularizers of the form

$$r_t(\boldsymbol{x}) \doteq \frac{\sigma_t}{2} \|\boldsymbol{x}\|^2, \forall t \geq 1.$$

The regularizers are set such that $r_{1:t}$ is 1-strongly convex w.r.t. the scaled Euclidean norm $\|\cdot\|_t \doteq \sqrt{\sigma_{1:t}} \|\cdot\|$ whose dual norm is $\|\cdot\|_{t,*} = 1/\sqrt{\sigma_{1:t}} \|\cdot\|$, hence we refer to $\sigma_t$ also as the "strong convexity" parameters.

Next, we select $\boldsymbol{g}_t^I$ according to the following: for $t = 1$, set $\boldsymbol{g}_1^I = -\boldsymbol{g}_1$ if $\epsilon_1 = 0$, and $\boldsymbol{g}_1^I = 0$ otherwise. For all $t \geq 2$:

$$\boldsymbol{g}_t^I = \begin{cases} -(\boldsymbol{p}_{1:t-1} + \tilde{\boldsymbol{g}}_t + \sigma_{1:t-1}\boldsymbol{x}_t) & \text{if } \boldsymbol{x}_t^{\text{uc}} \notin \mathcal{X} \\ 0 & \text{otherwise,} \end{cases} \quad (5)$$

where $\boldsymbol{x}_t^{\text{uc}}$ is the unconstrained iterate obtained by solving (4) in $\mathbb{R}^n$ (i.e., without the indicator function).

To see why we can always set $g_t^I$ as such, note that when $x_t^{\mathrm{uc}} \notin \mathcal{X}$, then $x_t \in \mathbf{bd}(\mathcal{X})$ (the projection of a point outside a compact convex set lies on the boundary). Hence, $\mathcal{N}_{\mathcal{X}}(x_t)$ contains vectors other than 0. In particular, it contains $-(p_{1:t-1} + \tilde{g}_t + \sigma_{1:t-1} x_t)$ and in fact all positive multiples of this vector. This follows directly from the optimality condition for the constrained iterate in (4), (see, e.g., (Beck, 2017, Thm. 3.67)), and the definition of the normal cone. For the second case, 0 is always a valid subgradient of the indicator function since, in this case, $x_t^{\mathrm{uc}} = x_t \in \mathcal{X}$.

Intuitively, this choice of linearization ensures an alternative state $p_{1:t}$ (instead of $g_{1:t}$). The former shall stop growing in norm compared to the latter beyond a certain $t$, since the action will hit the boundary of $\mathcal{X}$, starting the pruning thereafter. This will be formalized in the analysis, where we show that $\|p_{1:t}\|$ cannot grow faster than the regularization, which is added by us optimistically (i.e., $\propto \sqrt{E_t}$). This is not true in general for an arbitrary choice of linearization, particularly for the default choice of using only $g_t$.

We note that $g_t^I$ need not be non-zero at every time slot $t$ where $x_t^{\mathrm{uc}} \notin \mathcal{X}$. Instead, pruning of the accumulated state can be delayed for a fixed number of steps $k$, resulting in a *hybrid* state of the form $p_{1:k-1} + g_{k:t}$. This is discussed further in Appendix A.4.

In lines 10, 11, a prediction for the next cost is received, and the next unconstrained iterate is updated. Lastly, a feasible point is then recovered via a Euclidean[6] projection into $\mathcal{X}$.

Next, we characterize the dynamic regret of `OptFPRL` under different regularization strategies.

### 3.1. Dynamic Regret of **OptFPRL**

In this section, we explore different strategies for setting the regularization parameters $\sigma_t$, and analyze the resulting dynamic regret bounds. We begin by outlining the general setting shared across all regularization strategies.

**Settings 1.** Let $\mathcal{X} \subset \mathbb{R}^d$ be a compact, convex set such that $\|x\| \le R$ for all $x \in \mathcal{X}$. Let $\{f_t(\cdot), \tilde{f}_t(\cdot)\}_{t=1}^T$ be any sequence of $L$-Lipschitz convex functions. Define the path length $P_T$ as in (1), the cumulative prediction error $E_T$ as in (2), and the hybrid term $H_T \doteq \sum_{t=1}^{T-1} \epsilon_t \|u_{t+1} - u_t\|$.

#### 3.1.1. $P_T$-AGNOSTIC REGULARIZATION

The first regularization strategy we consider is the standard optimistic one, which sets $\sigma_t$ such that $\sigma_{1:t} \propto \sqrt{E_T}$:

$$
\sigma = \frac{1}{4R}, \quad \sigma_1 = \sigma \epsilon_1,
$$
$$
\sigma_t = \sigma \left( \sqrt{E_t} - \sqrt{E_{t-1}} \right), \forall t \ge 2. \tag{6}
$$

---

[6]Technically, the projection is w.r.t. $\|\cdot\|_t$. Since $\|\cdot\|_t$ is a scaled Euclidean, the result is the same.

**Theorem 3.1.** *Under Settings 1, Alg. 1 run with the regularization strategy in* (6) *produces points* $\{x_t\}_{t=1}^T$ *such that, for any $T$, the dynamic regret $\mathcal{R}_T$ satisfies:*

$$
\mathcal{R}_T \le \left( 5.8R + (1/2)P_T \right) \sqrt{E_T} + H_T \tag{7}
$$
$$
= \mathcal{O} \left( (1 + P_T) \sqrt{E_T} \right).
$$

**Remarks.** We observe from (7) that all terms in the regret bound are modulated by the prediction errors. When the predictions are perfect, the bound reduces to zero (regardless of $P_T$), and it gracefully degrades as prediction errors grow. Note that by Cauchy-Schwarz and the boundness of $\mathcal{X}$, the hybrid term can be bounded as $H_T \le \sqrt{2R}\sqrt{E_T P_T}$. Therefore, the overall bound is never worse than $\mathcal{O}((1 + P_T)\sqrt{E_T})$, matching known OMD results when $P_T$ is not accounted for, and tighter when predictions are accurate. In the static comparator case ($P_T = 0$), the bound recovers the standard $\mathcal{O}(\sqrt{E_T})$ result.

#### 3.1.2. REGULARIZATION WITH PRIOR $P_T$ KNOWLEDGE

In many cases, $P_T$ can be provided to the algorithm a priori as a measure of the comparator's complexity (e.g., (Si-Salem et al., 2024; Yang et al., 2016; Besbes et al., 2015)). That is, we wish to compete against any sequence whose path length is at most the provided value of $P_T$. With this given target, we can adjust the regularization to account for the expected nonstationarity and obtain better bounds, as outlined next:

$$
\sigma = \frac{1}{2\sqrt{2RP_T'}}, \quad P_T' \doteq 2R + P_T, \quad \sigma_1 = \sigma \epsilon_1,
$$
$$
\sigma_t = \sigma \left( \sqrt{E_t} - \sqrt{E_{t-1}} \right), \forall t \ge 2. \tag{8}
$$

**Theorem 3.2.** *Under Settings 1, Alg. 1 run with the regularization strategy in* (8) *produces points* $\{x_t\}_{t=1}^T$ *such that, for any $T$, the dynamic regret $\mathcal{R}_T$ satisfies:*

$$
\mathcal{R}_T \le \left( 4\sqrt{2R^2 + P_T} + \frac{R}{8} + \sqrt{\frac{RP_T}{2}} \right) \sqrt{E_T} + H_T
$$
$$
= \mathcal{O} \left( (1 + \sqrt{P_T}) \sqrt{E_T} \right).
$$

**Remarks.** This regularization strategy preserves the full modulation by prediction errors while also matching the minimax-optimal bound $R_T = \Omega(\sqrt{(1 + P_T)T})$ (Zhang et al., 2018, Thm. 2) even when all predictions fail. This type of bound is new for FTRL-style algorithms. Furthermore, in its full dependency on $E_T$ (i.e., without $P_T$ or constant terms that are independent of $E_T$), it represents a refinement even compared to OMD. More broadly, access to a prior bound on $P_T$ enables tailoring the regularization to the expected nonstationarity, allowing us to safeguard the minimax rate while still adapting to prediction accuracy.

### 3.1.3. REGULARIZATION WITH UNKNOWN $P_T$

If $P_T$ is unknown but observable, it can be estimated online alongside $E_t$. However, since $\sqrt{E_t/P_t}$ is no longer necessarily monotonic, we should safeguard against negative regularization coefficients. First, define the augmented seen path length at $t$ as:

$$P'_t \doteq 2R + P_t = 2R + \sum_{\tau=1}^{t-1} \|\boldsymbol{u}_{\tau+1} - \boldsymbol{u}_\tau\|.$$

We adopt a regularization strategy that attempts to track $\sqrt{E_T/P_T}$ by its online estimate $\sqrt{E_t/P_t}$, while using a $\max(\cdot, \cdot)$ operator to ensure non-negative regularization.

$$
\sigma = \frac{1}{2\sqrt{2R}}, \quad \sigma_1 = \frac{\sigma \epsilon_1}{\sqrt{P'_1}},
$$
$$
\sigma_t = \sigma \max\left(0, \sqrt{\frac{E_t}{P'_t}} - \sqrt{\frac{E_{t-1}}{P'_{t-1}}}\right), \forall t \geq 2. \tag{9}
$$

**Theorem 3.3.** *Under Settings 1, Alg. 1 run with the regularization strategy in (9) produces points $\{\boldsymbol{x}_t\}_{t=1}^T$ such that, for any $T$ the dynamic regret $\mathcal{R}_T$ satisfies:*

$$
\mathcal{R}_T \leq 5.5\sqrt{R}\sqrt{E_T P'_T} + H_T + \sqrt{R/2}\, A_T
$$
$$
= \mathcal{O}\left((1 + \sqrt{P_T})\sqrt{E_T} + A_T\right), \text{where}
$$
$$
A_T \doteq \sum_{t=1}^T \sum_{\tau \in [t]^+} \left(\sqrt{\frac{E_{\tau-1}}{P'_{\tau-1}}} - \sqrt{\frac{E_\tau}{P'_\tau}}\right)\|\boldsymbol{u}_{t+1} - \boldsymbol{u}_t\|,
$$
$$
[t]^+ = \left\{2 \leq \tau \leq t \;\middle|\; \sqrt{\frac{E_{\tau-1}}{P'_{\tau-1}}} - \sqrt{\frac{E_\tau}{P'_\tau}} \geq 0\right\}.
$$

**Remarks.** Note that this bound resembles that of Theorem 3.2, except for the additional term $A_T$, which arises due to the potential non-monotonicity of the estimated quantity $\sqrt{E_t/P_t}$. If this sequence were monotonic and non-decreasing, $A_T$ would vanish since $[t]^+$ would be empty. This case allows us to recover the bound of Theorem 3.2 without requiring prior knowledge of $P_T$. In contrast, in the worst-case scenario, where the sequence alternates direction at every round, we obtain $A_T = \mathcal{O}(\sqrt{E_T}(P_T + 1))$ (see Appendix C.3.1), leading to the looser bound in Theorem 3.1. A comparable correction term to $A_T$ also appears in the optimistic OMD framework (Scroccaro et al., 2023, Remark 2.19), and likewise depends on monotonicity, reaching $\sqrt{E_T}P_T$ in the worst case. The above-mentioned bound is, however, more interpretable.

A workaround to this monotonicity issue is a doubling-trick variant (Jadbabaie et al., 2015), which, however, introduces a slight problem-independence through a multiplicative $\Theta(\log T)$ factor, and an additive $\Theta(\log T\sqrt{P_T})$ factor that persists even under perfect predictions.

### 3.1.4. RECURSIVE REGULARIZATION

In this subsection, we employ the regularization strategy of AdaFTRL (Orabona & Pál, 2018), which sets the strong convexity parameters recursively, ensuring the minimal required regularization. Namely, we do not set $\sigma_t$ such that $\sigma_{1:t} \propto \sqrt{E_t}$, as in the strategies discussed earlier, and prior works on optimistic dynamic regret. Instead, we set $\sigma_t \propto \delta_t \doteq h_{0:t-1}(\boldsymbol{x}_t) + \langle \boldsymbol{p}_t, \boldsymbol{x}_t \rangle - \min_{\boldsymbol{x}}(h_{0:t-1}(\boldsymbol{x}) + \langle \boldsymbol{p}_t, \boldsymbol{x} \rangle)$, where $h_t(\cdot)$ is the regularized loss: $h_t(\cdot) = \langle \boldsymbol{p}_t, \cdot \rangle + r_t(\cdot)$. This choice of $\sigma_t$ is such that $\sigma_{1:t} \propto c \leq \sqrt{E_t}$.

In other words, we increase the strong convexity exactly in proportion to the (regularized) cumulative loss observed at $t$ (denoted as $\delta_t$), rather than an upper bound on that loss. Since the added strong convexity essentially determines the regret bound, this leads to tighter bounds overall that are no worse than the ones derived earlier. Namely, we use the following regularization strategy:

$$
\sigma = \frac{1}{8R^2}, \quad \sigma_t = \sigma \delta_t, \quad \delta_1 = \langle \boldsymbol{g}_1, \boldsymbol{x}_1 \rangle - \min_{\boldsymbol{x} \in \mathcal{X}} \langle \boldsymbol{g}_1, \boldsymbol{x} \rangle
$$
$$
\delta_t = h_{0:t-1}(\boldsymbol{x}_t) + \langle \boldsymbol{p}_t, \boldsymbol{x}_t \rangle \tag{10}
$$
$$
- \min_{\boldsymbol{x} \in \mathcal{X}}(h_{0:t-1}(\boldsymbol{x}) + \langle \boldsymbol{p}_t, \boldsymbol{x} \rangle), \quad \forall t \geq 2.
$$

**Theorem 3.4.** *Under Settings 1, Alg. 1 run with the regularization strategy in (10) produces points $\{\boldsymbol{x}_t\}_{t=1}^T$ such that, for any $T$, the dynamic regret $\mathcal{R}_T$ satisfies:*

$$
\mathcal{R}_T \leq 1.1\, \delta_{1:T} + \sum_{t=1}^{T-1} \frac{1}{4R}\delta_{1:t}\|\boldsymbol{u}_{t+1} - \boldsymbol{u}_t\| + H_T
$$
$$
\leq (3.7R + P_T)\sqrt{E_T} + H_T = \mathcal{O}\left((1 + P_T)\sqrt{E_T}\right).
$$

**Remarks.** Since the $\delta_t$ terms are often smaller than their upper bound, this minimal regularization approach is particularly advantageous in dynamic environments, where regularization terms sum is nested within the primary sum over $[T]$. While this tuning strategy has been employed previously in *static* settings for optimistic learning (Flaspohler et al., 2021), it has not yet been leveraged for optimism in dynamic settings[7]. Nonetheless, the recursive nature of this regularization adds a technical challenge since the accumulated strong convexity does not have a closed-form expression in terms of $\sqrt{E_t}$. Fortunately, this recursion is still shown to be bounded by $\mathcal{O}(\sqrt{E_t})$ via tools developed in the AdaFTRL framework.

---

[7]We note, however, the related temporal-variation-based dynamic regret bound in Implicit OMD (Campolongo & Orabona, 2021), where the structure of "function changes", rather than "prediction errors", is exploited.

## 4. Tools for FTRL's Dynamic Regret Analysis

In this section, we present the primary analytical tools used to derive the regret bounds of the previous section. These results extend traditional FTRL analyses to incorporate both optimism and dynamic comparators. First, we bound the regret via linearization:

$$\mathcal{R}_T = \sum_{t=1}^{T} j_t(\boldsymbol{x}_t) - j_t(\boldsymbol{u}_t) \leq \sum_{t=1}^{T} \langle \boldsymbol{p}_t, \boldsymbol{x}_t - \boldsymbol{u}_t \rangle,$$

where $j_t(\boldsymbol{x}) \doteq f_t(\boldsymbol{x}) + I_{\mathcal{X}}(\boldsymbol{x})$. The inequality follows directly from the convexity of $j_t(\cdot)$,[8] noting that $\boldsymbol{p}_t \in \partial j_t(\boldsymbol{x}_t)$ by definition of $\boldsymbol{p}_t$ (recall $\boldsymbol{p}_t = \boldsymbol{g}_t + \boldsymbol{g}_t^I$, with $\boldsymbol{g}_t \in \partial f_t(\boldsymbol{x}_t), \boldsymbol{g}_t^I \in \partial I_{\mathcal{X}}(\boldsymbol{x}_t))$.

Our analysis begins with a generalization of the main FTRL lemma (McMahan, 2017, Lem. 5).

**Lemma 4.1.** *(Strong Dynamic Optimistic FTRL).* *Let* $\{f_t(\cdot), \tilde{f}_t(\cdot), \boldsymbol{u}_t\}_{t=1}^{T}$ *be an arbitrary set of functions, predicted functions, and comparators within* $\mathcal{X}$, *respectively. Let* $r_t(\cdot)$ *be non-negative regularization functions such that*

$$\boldsymbol{x}_{t+1} \doteq \operatorname*{argmin}_{\boldsymbol{x}} h_{0:t}(\boldsymbol{x}) + \tilde{f}_{t+1}(\boldsymbol{x})$$

*is well-defined, where* $h_0(\boldsymbol{x}) \doteq I_{\mathcal{X}}(\boldsymbol{x})$, *and* $\forall t \geq 1$:

$$h_t(\boldsymbol{x}) \doteq \langle \boldsymbol{p}_t, \boldsymbol{x} \rangle + r_t(\boldsymbol{x}), \quad \boldsymbol{p}_t \in \partial j_t(\boldsymbol{x}_t).$$

*Then, the algorithm that selects the actions* $\boldsymbol{x}_{t+1}, \forall t$ *achieves the following dynamic regret bound:*

$$\mathcal{R}_T \leq \sum_{t=1}^{T} \overbrace{h_{0:t}(\boldsymbol{x}_t) - h_{0:t}(\boldsymbol{x}_{t+1}) - r_t(\boldsymbol{x}_t)}^{(I)}$$
$$+ \sum_{t=1}^{T-1} \underbrace{h_{0:t}(\boldsymbol{u}_{t+1}) - h_{0:t}(\boldsymbol{u}_t)}_{(II)} + r_t(\boldsymbol{u}_t).$$

The regret is thus decomposed into *three* main parts. Part **(I)** measures the penalty incurred due to not knowing $\boldsymbol{g}_t$ when deciding $\boldsymbol{x}_t$. The second part **(II)** measures the penalty incurred due to the non-stationarity of the environment (change of comparators). The last part, $r_t(\boldsymbol{u}_t)$, is a user-influenced quantity that reflects the amount of regularization introduced and will be traded off against the other terms in the bound that benefit from more regularization.

Next, we describe the upper bounds:

$$(I) \leq \min\left(\frac{1}{2}\|\boldsymbol{g}_t - \tilde{\boldsymbol{g}}_t\|_{t-1,*}^2, 2R\epsilon_t\right)$$

$$(II) \leq (\|\boldsymbol{p}_{1:t}\| + R\sigma_{1:t})\|\boldsymbol{u}_{t+1} - \boldsymbol{u}_t\|$$

[8]see, e.g., (Shalev-Shwartz, 2012, Sec. 2.4), for more details on linearization.

**(I)** is a result of a generalization of a common tool in OCO which bounds the difference in the value of a strongly convex function ($h_{0:t}(\cdot)$) when evaluated at a "partial"[9] minimizer ($\boldsymbol{x}_t$) versus the global minimizer (say, $\boldsymbol{y}$, and hence $\boldsymbol{x}_{t+1}$ also), which we state below.

**Lemma 4.2.** *Let each function* $h_{0:t}(\cdot)$ *be 1-strongly convex w.r.t. a norm* $\|\cdot\|_t$ *defined as in Lemma 4.1. Let* $\boldsymbol{x}_t = \operatorname{argmin}_{\boldsymbol{x}} h_{0:t-1}(\boldsymbol{x}) + \tilde{f}_t(\boldsymbol{x})$. *Then, we have the inequality*

$$h_{0:t}(\boldsymbol{x}_t) - h_{0:t}(\boldsymbol{x}_{t+1}) - r_t(\boldsymbol{x}_t) \leq \frac{1}{2}\|\boldsymbol{g}_t - \tilde{\boldsymbol{g}}_t\|_{t-1,*}^2.$$

The second term of the min is in fact a crude regularization-independent bound on the per-slot loss of $\boldsymbol{x}_t$ compared to the omniscient $\boldsymbol{x}_{t+1}$. We detail both bounds in Appendix B.

**(II)** follows directly from the first-order inequality for convex functions applied to $h_{0:t}(\cdot)$. Note that it is exactly the $\|\boldsymbol{p}_{1:t}\|$ term that explains the potential failure of FTRL in dynamic environments, even with a *constant* path length. Specifically, a trivial bound on this norm is linear in $t$ and hence is super-linear in $T$ even with one switch in the comparators. This is precisely the "vulnerability" that is exploited in the impossibility result in (Jacobsen & Cutkosky, 2022, Thm. 2). Next, we show how pruning can *tie* this $\|\boldsymbol{p}_{1:t}\|$ term to the regularization parameters $\sigma_{1:t}$ (which we control), thus ensuring its sub-linearity.

**Lemma 4.3.** *(Optimistically Bounded State) Let* $\{\boldsymbol{p}_t\}_{t=1}^{T}$ *be a sequence of vectors such that each* $\boldsymbol{p}_t = \boldsymbol{g}_t + \boldsymbol{g}_t^I$, *where* $\boldsymbol{g}_t \in \partial f_t(\boldsymbol{x}_t)$, *and* $\boldsymbol{g}_t^I \in \partial I_{\mathcal{X}}(\boldsymbol{x}_t)$ *is chosen according to the construction in* (5). *That is,* $\boldsymbol{p}_t$ *corresponds to the linearization of the composite function* $f_t(\cdot) + I_{\mathcal{X}}(\cdot)$ *around* $\boldsymbol{x}_t$. *Then, for any* $t$, *the following holds:*

$$\|\boldsymbol{p}_{1:t}\| \leq R\sigma_{1:t-1} + \epsilon_t.$$

*Proof.* First, we begin by showing that

$$\text{For any } t, \boldsymbol{x}_t^{\text{uc}} \in \mathcal{X} \implies \|\boldsymbol{p}_{1:t}\| \leq \sigma_{1:t-1}R + \epsilon_t. \quad (11)$$

Since $\boldsymbol{x}_t^{\text{uc}} \in \mathcal{X}$, we know that $\boldsymbol{x}_t = \Pi(\boldsymbol{x}_t^{\text{uc}}) = \boldsymbol{x}_t^{\text{uc}}$ (The projection of a point within the set is itself), and hence $\boldsymbol{x}_t$ is a minimizer of the unconstrained update rule too:

$$\boldsymbol{x}_t = \operatorname*{argmin}_{\boldsymbol{x} \in \mathbb{R}^n} \langle \boldsymbol{p}_{1:t-1}, \boldsymbol{x} \rangle + r_{1:t-1}(\boldsymbol{x}) + \tilde{f}_t(\boldsymbol{x}) \quad (12)$$

From the first-order optimality condition for unconstrained problems (e.g., (Beck, 2017, Thm. 3.63)), we know that 0 is an element of the subdifferential of (12) at $\boldsymbol{x}_t$. Thus, $\exists \tilde{\boldsymbol{g}}_t \in \partial f_t(\boldsymbol{x}_t)$ such that

$$0 = \boldsymbol{p}_{1:t-1} + \sigma_{1:t-1}\boldsymbol{x}_t + \tilde{\boldsymbol{g}}_t$$
$$\implies \boldsymbol{p}_{1:t-1} = -\sigma_{1:t-1}\boldsymbol{x}_t - \tilde{\boldsymbol{g}}_t, \quad (13)$$

[9]Recall that $\boldsymbol{x}_t$ does not minimize $h_{0:t}$, but a related function.

and we have that the norm of the state $\boldsymbol{p}_{1:t}$ satisfies

$$\|\boldsymbol{p}_{1:t}\| = \|\boldsymbol{p}_{1:t-1} + \boldsymbol{p}_t\|$$
$$\overset{(a)}{=} \|\boldsymbol{p}_{1:t-1} + \boldsymbol{g}_t\| \overset{(b)}{=} \| -\sigma_{1:t-1}\boldsymbol{x}_t - \tilde{\boldsymbol{g}}_t + \boldsymbol{g}_t\|$$
$$\leq \| -\sigma_{1:t-1}\boldsymbol{x}_t\| + \|\boldsymbol{g}_t - \tilde{\boldsymbol{g}}_t\|$$
$$= \sigma_{1:t-1}\|\boldsymbol{x}_t\| + \epsilon_t \overset{(c)}{\leq} \sigma_{1:t-1}R + \epsilon_t,$$

where $(a)$ holds because $\boldsymbol{g}_t^I = 0$ (from (5) & $\boldsymbol{x}_t^{\text{uc}} \in \mathcal{X}$), $(b)$ from (13), and $(c)$ from the set size.

Next, we show that when $\boldsymbol{x}_t^{\text{uc}} \notin \mathcal{X}$, the state is forced via pruning to be bounded:

For any $t$, $\boldsymbol{x}_t^{\text{uc}} \notin \mathcal{X} \implies \|\boldsymbol{p}_{1:t}\| \leq \sigma_{1:t-1}R + \epsilon_t.$   (14)

When $\boldsymbol{x}_t^{\text{uc}} \notin \mathcal{X}$, the pruning condition is activated, which ensures that

$$\boldsymbol{p}_{1:t} = \boldsymbol{p}_{1:t-1} + \overbrace{\boldsymbol{g}_t \underbrace{-\boldsymbol{p}_{1:t-1} - \tilde{\boldsymbol{g}}_t - \sigma_{1:t-1}\boldsymbol{x}_t}_{\boldsymbol{g}_t^I}}^{\boldsymbol{p}_t}$$
$$= \boldsymbol{g}_t - \tilde{\boldsymbol{g}}_t - \sigma_{1:t-1}\boldsymbol{x}_t.$$
Thus, $\|\boldsymbol{p}_{1:t}\| = \|\boldsymbol{g}_t - \tilde{\boldsymbol{g}}_t - \sigma_{1:t-1}\boldsymbol{x}_t\|$
$$\leq \|\boldsymbol{g}_t - \tilde{\boldsymbol{g}}_t\| + \sigma_{1:t-1}\|\boldsymbol{x}_t\|$$
$$\leq \epsilon_t + \sigma_{1:t-1}R.$$

The lemma statement follows by (11) and (14).   □

### 4.1. Regret Bound Derivation

We show the proof of Theorem 3.1 since it illustrates how the presented tools come together to obtain the results. It also provides a sufficient basis for sketching the proofs of the remaining theorems.

*Proof of Theorem 3.1.* We begin from the main lemma, Lemma 4.1, with the bounds on **(I)** and **(II)** terms:

$$\mathcal{R}_T \leq \sum_{t=1}^{T}\left(\min\left(\frac{1}{2}\|\boldsymbol{g}_t - \tilde{\boldsymbol{g}}_t\|_{t-1,*}^2, 2R\epsilon_t\right) + r_t(\boldsymbol{u}_t)\right)$$
$$+ \sum_{t=1}^{T-1}\left((R\sigma_{1:t} + \|\boldsymbol{p}_{1:t}\|)\|\boldsymbol{u}_{t+1} - \boldsymbol{u}_t\|\right) \quad (15)$$
$$\overset{(a)}{\leq} \sum_{t=1}^{T}\min\left(\frac{1}{2}\|\boldsymbol{g}_t - \tilde{\boldsymbol{g}}_t\|_{t-1,*}^2, 2R\epsilon_t\right) + \frac{R^2}{2}\sigma_{1:T}$$
$$+ \sum_{t=1}^{T-1}\left((2R\sigma_{1:t} + \epsilon_t)\|\boldsymbol{u}_{t+1} - \boldsymbol{u}_t\|\right)$$
$$= \sum_{t=1}^{T}\min\left(\frac{\epsilon_t^2}{2\sigma\sqrt{E_{t-1}}}, 2R\epsilon_t\right) + \frac{R^2}{2}\sigma\sqrt{E_T}$$
$$+ \sum_{t=1}^{T-1}\left(\left(2R\sigma\sqrt{E_t} + \epsilon_t\right)\|\boldsymbol{u}_{t+1} - \boldsymbol{u}_t\|\right) \quad (16)$$

$$\overset{(b)}{\leq} 4\sqrt{2}R\sqrt{E_T} + \frac{R}{8}\sqrt{E_T}$$
$$+ \frac{1}{2}\sum_{t=1}^{T-1}\sqrt{E_t}\|\boldsymbol{u}_{t+1} - \boldsymbol{u}_t\| + H_T$$
$$\overset{(c)}{\leq} 5.8R\sqrt{E_T} + \frac{\sqrt{E_T}}{2}P_T + H_T$$

where $(a)$ follows from the boundedness of $\mathcal{X}$, the state bound in Lemma 4.3, and the fact that $\sigma_{1:t-1} \leq \sigma_{1:t}$; the equality follows from the definition of $\|\cdot\|_{*,t}$ and the telescoping structure of $\sigma_{1:t}$; $(b)$ follows from applying Lemma D.3 to the $\sum_t \min(\cdot, \cdot)$ term, which bounds the sum of the non-increasing quantities, and substituting the expression for $\sigma$; $(c)$ uses that $\sqrt{E_t}$ is non-decreasing.  □

Overall, after using the results developed in Sec. 4, obtaining the exact results of the theorems mainly hinges on available tools from the OCO literature on bounding the sum of non-increasing functions.

**Proof sketch of the other theorems.** The proofs of the other theorems follow similar main steps, and are detailed in the Appendix. When the $\sigma$ parameter is normalized by $\sqrt{P_T}$, it can be seen from (16) that the $P_T$ term will also be divided by the same, recovering the result of Theorem 3.2.

For Theorem 3.3, we write the sum $\sigma_{1:t}$ in (15) as the non-monotone $\sqrt{E_t/P_t'} - \sqrt{E_{t-1}/P_{t-1}'}$ provided that we add the corrective term $A_T$ defined earlier. The former tracks the desired quantity $\sqrt{E_T/P_T'}$, which is what is required (and was used in Theorem 3.2) to recover the $\sqrt{E_T P_T}$ dependence. The corrective sum $A_T$ is handled independently.

Lastly, for Theorem 3.4, we start again from Lemma 4.1, but instead of using the upper bound of the term **(I)**, we use the $\delta_t$ terms. This results in a recursion whose solution is fortunately available among AdaFTRL lemmas.

## 5. Conclusion

This paper introduced an optimistic FTRL variant with new data-dependent dynamic regret guarantees, extending classical FTRL results and advancing our understanding of this foundational framework in non-stationary environments. These bounds are explicitly tied to the accuracy of predictions, offering refined performance guarantees in dynamic settings. The gist of our proposal lies in a simple pruning rule that modulates the memory (or state) of FTRL based on the quality of the obtained decisions, preventing the accumulation of redundant (negative) gradients that align with iterates on the boundary of the decision set. This technique can be extended to more flexible pruning strategies that control pruning magnitude or introduce additional pruning conditions, enabling a spectrum of algorithms ranging from fully "lazy" to fully "agile" iterates.

## Acknowledgements

This work was supported by the Dutch National Growth Fund project "Future Network Services", and by the European Commission through Grants No. 101139270 "ORIGAMI" and No. 101192462 "FLECON-6G".

## Impact Statement

This paper presents work whose goal is to advance the field of Machine Learning. There are many potential societal consequences of our work, none which we feel must be specifically highlighted here.

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

# Contents

## A. `OptFPRL`

Below, we restate the main ingredients of `OptFPRL` using a more detailed presentation.

### A.1. Update Rule

Recall the notation $\boldsymbol{a}_{1:t} \doteq \sum_{\tau=1}^{t} \boldsymbol{a}_\tau$. The update rule of `OptFPRL` is:

$$\boldsymbol{x}_{t+1} = \operatorname*{argmin}_{\boldsymbol{x}} \{ \langle \underbrace{\boldsymbol{p}_{1:t}}_{\text{State vector}}, \boldsymbol{x} \rangle + \underbrace{r_{1:t}(\boldsymbol{x})}_{\text{Incremental regularizers}} + \underbrace{\tilde{f}_{t+1}(\boldsymbol{x})}_{\text{Prediction}} + \underbrace{I_{\mathcal{X}}(\boldsymbol{x})}_{\text{Set constraint}} \}. \tag{17}$$

where each $\boldsymbol{p}_t$ is the sum of the linearization of the cost function $f_t(\boldsymbol{x}_t)$, and some choice from the cone $\mathcal{N}_{\mathcal{X}}(\boldsymbol{x}_t)$.

$$\boldsymbol{p}_t = \boldsymbol{g}_t + \boldsymbol{g}_t^I, \ \ \boldsymbol{g}_t \in \partial f_t(\boldsymbol{x}_t), \ \ \boldsymbol{g}_t^I \in \partial I_{\mathcal{X}}(\boldsymbol{x}_t) = \mathcal{N}_{\mathcal{X}}(\boldsymbol{x}_t). \tag{18}$$

### A.2. Regularizer

We use the scaled Euclidean regularizer,

$$r_t(\boldsymbol{x}) = \frac{\sigma_t}{2} \|\boldsymbol{x}\|^2,$$

where $\sigma_t$ is the strong convexity parameter that will be set in every version of the algorithm. Note that the sum $r_{1:t}(\cdot)$ is $\sigma_{1:t}$-strongly convex with respect to $\|\cdot\|$, or equivalently 1-strongly convex with respect to $\|\cdot\|_t \doteq \sqrt{\sigma_{1:t}} \|\cdot\|$.

### A.3. Pruning

Here, we present a mechanism for selecting the vectors $\boldsymbol{g}_t^I$ from the cone $\mathcal{N}_{\mathcal{X}}(\boldsymbol{x}_t)$. Define the unconstrained iterate $\boldsymbol{x}_t^{\text{uc}}$, which is obtained by solving the update rule without the indicator function

$$\boldsymbol{x}_{t+1}^{\text{uc}} = \operatorname*{argmin}_{\boldsymbol{x}} \langle \boldsymbol{p}_{1:t}, \boldsymbol{x} \rangle + r_{1:t}(\boldsymbol{x}) + \tilde{f}_{t+1}(\boldsymbol{x}).$$

Pruning will depend on whether this iterate belongs to the set $\mathcal{X}$ or not.

The unconstrained iterate $\boldsymbol{x}_t^{\text{uc}}$ always exists and is unique due to the strong convexity of $r_{1:t}(\cdot)$. However, its membership in $\mathcal{X}$ depends on the exact degree of strong convexity. When $\sigma_{1:t} = 0$, we consider $\boldsymbol{x}_t^{\text{uc}} \notin \mathcal{X}$, as the minimizer of an unconstrained linear function does not exist.

We select $\boldsymbol{g}_t^I$ as follows: for $t = 1$, set $\boldsymbol{g}_1^I = -\boldsymbol{g}_1$ if $\epsilon_1 = \sigma_1 = 0$, and $\boldsymbol{g}_1^I = 0$ otherwise. For all $t \geq 2$:

$$\boldsymbol{g}_t^I = \begin{cases} -(\boldsymbol{p}_{1:t-1} + \tilde{\boldsymbol{g}}_t + \sigma_{1:t-1}\boldsymbol{x}_t) & \text{if } \boldsymbol{x}_t^{\text{uc}} \notin \mathcal{X} \\ 0 & \text{otherwise.} \end{cases} \tag{19}$$

Recall that this is a valid choice because when $\boldsymbol{x}_t^{\text{uc}} \notin \mathcal{X}$, then $\boldsymbol{x}_t \in \mathbf{bd}(\mathcal{X})$. Hence, $\mathcal{N}_{\mathcal{X}}(\boldsymbol{x}_t)$ contains elements other than 0, and in particular $-(\boldsymbol{p}_{1:t-1} + \tilde{\boldsymbol{g}}_t + \sigma_{1:t-1}\boldsymbol{x}_t)$. This is a direct result of the optimality condition for the constrained iterate (17), see, e.g., (Beck, 2017, Thm. 3.67). For the second case, 0 is always a valid subgradient of the indicator function since in this case $\boldsymbol{x}_t^{\text{uc}} = \boldsymbol{x}_t \in \mathcal{X}$.

Due to the FTRL/OMD connections, detailed in the related work, the update rule can be made equivalent to the "one-step optimistic MD" formulation in (Joulani et al., 2020), under the consistent pruning in (19) using the flexible $q_t(\cdot)$ terms in their Ada-MD update.

### A.4. Dual-perspective

To provide a dual view of the proposed FTRL variant, we look at the update step through the lens of dual maps.

First, recall the definition of the convex conjugate $r^\star(\cdot)$ of a closed convex function $r(\cdot)$:

$$r^\star(\boldsymbol{y}) \doteq \sup_{\boldsymbol{x}} \langle \boldsymbol{x}, \boldsymbol{y} \rangle - r(\boldsymbol{x}).$$

The update of FTRL can be expressed using the above definition applied to a potentially time-varying $r(\cdot)$. Namely, the standard FTRL update can be expressed as:

$$\boldsymbol{x}_{t+1}^{\text{RL}} = \arg\min_x \langle \boldsymbol{g}_{1:t}, \boldsymbol{x} \rangle + r_{0:t}(\boldsymbol{x}) = \nabla r_{0:t}^\star(-\boldsymbol{g}_{1:t}),$$

where $r_{0:t}^\star(\cdot)$ is the conjugate of the cumulative regularizer $r_{1:t}(\cdot)$, restricted to $\mathcal{X}$ via $r_0 = I_{\mathcal{X}}$. From this viewpoint, FTRL maintains the state as cumulative gradients in dual space.

The update of `OptFPRL` can be interpreted as:

$$\boldsymbol{x}_{t+1} = \arg\min_x \langle \boldsymbol{p}_{1:t}, \boldsymbol{x} \rangle + r_{0:t}(\boldsymbol{x}) = \nabla r_{0:t}^\star(-\boldsymbol{p}_{1:t}) = \nabla r_{0:t}^\star(\nabla r_{1:t-k-1}(\boldsymbol{x}_{t-k}) - \boldsymbol{g}_{t-k:t}),$$

where $t - k$ denotes the most recent step at which we chose to prune. The last equality holds by the definition of $\boldsymbol{g}_k^I$ (assuming no predictions). Intuitively, we retain explicit gradient history only since the last pruning step $t - k$, while summarizing earlier history implicitly via the dual mapping of $\boldsymbol{x}_{t-k}$. The crux of the paper is showing that the way history is split (explicitly tracked after pruning, and implicitly captured before) is what controls dynamic regret.

### A.5. Regret Characterization

Define the dynamic regret metric against any set of comparators $\{\boldsymbol{u}_t\}_t$ as:

$$\mathcal{R}_T \doteq \sum_{t=1}^{T} f_t(\boldsymbol{x}_t) - f_t(\boldsymbol{u}_t) = \sum_{t=1}^{T} j_t(\boldsymbol{x}_t) - j_t(\boldsymbol{u}_t) \leq \sum_{t=1}^{T} \langle \boldsymbol{p}_t, \boldsymbol{x}_t - \boldsymbol{u}_t \rangle, \tag{20}$$

where $j_t(\boldsymbol{x}) \doteq f_t(\boldsymbol{x}) + I_{\mathcal{X}}(\boldsymbol{x})$. The inequality holds by the linearization principle of convex functions (Shalev-Shwartz, 2012, Sec. 2.4), noticing that we indeed have $\boldsymbol{p}_t \in \partial j_t(\boldsymbol{x}_t)$ due the definition of $\boldsymbol{p}_t$ in (18), and the fact that the subdifferential of the sum contains the sum of the subdifferentials (e.g., (Orabona, 2022, Thm. 22)).

## B. Missing Proofs for Section 4

### B.1. The Strong Dynamic Optimistic FTRL Lemma

**Lemma 4.1.** *(Strong Dynamic Optimistic FTRL). Let $\{f_t(\cdot), \tilde{f}_t(\cdot), \boldsymbol{u}_t\}_{t=1}^{T}$ be an arbitrary set of functions, predicted functions, and comparators within $\mathcal{X}$, respectively. Let $r_t(\cdot)$ be non-negative regularization functions such that*

$$\boldsymbol{x}_{t+1} \doteq \underset{\boldsymbol{x}}{\operatorname{argmin}}\, h_{0:t}(\boldsymbol{x}) + \tilde{f}_{t+1}(\boldsymbol{x})$$

*is well-defined, where $h_0(\boldsymbol{x}) \doteq I_{\mathcal{X}}(\boldsymbol{x})$, and $\forall t \geq 1$:*

$$h_t(\boldsymbol{x}) \doteq \langle \boldsymbol{p}_t, \boldsymbol{x} \rangle + r_t(\boldsymbol{x}), \quad \boldsymbol{p}_t \in \partial j_t(\boldsymbol{x}_t).$$

*Then, the algorithm that selects the actions $\boldsymbol{x}_{t+1}, \forall t$ achieves the following dynamic regret bound:*

$$\mathcal{R}_T \leq \sum_{t=1}^{T} \overbrace{h_{0:t}(\boldsymbol{x}_t) - h_{0:t}(\boldsymbol{x}_{t+1}) - r_t(\boldsymbol{x}_t)}^{(I)} + \sum_{t=1}^{T-1} \underbrace{h_{0:t}(\boldsymbol{u}_{t+1}) - h_{0:t}(\boldsymbol{u}_t)}_{(II)} + r_t(\boldsymbol{u}_t).$$

*Proof.*

$$\sum_{t=1}^{T} h_t(\boldsymbol{x}_t) - \sum_{t=1}^{T} h_t(\boldsymbol{u}_t) - \tilde{f}_{T+1}(\boldsymbol{u}_T)$$

$$= \sum_{t=1}^{T} (h_{0:t}(\boldsymbol{x}_t) - h_{0:t-1}(\boldsymbol{x}_t)) - \left( \sum_{t=1}^{T} (h_{0:t}(\boldsymbol{u}_t) - h_{0:t-1}(\boldsymbol{u}_t)) + \tilde{f}_{T+1}(\boldsymbol{u}_T) \right)$$

$$= \sum_{t=1}^{T} h_{0:t}(\boldsymbol{x}_t) - \sum_{t=1}^{T} h_{0:t-1}(\boldsymbol{x}_t) - \left( \sum_{t=1}^{T} h_{0:t}(\boldsymbol{u}_t) + \tilde{f}_{T+1}(\boldsymbol{u}_T) - \sum_{t=1}^{T} h_{0:t-1}(\boldsymbol{u}_t) \right)$$

$$= \sum_{t=1}^{T} h_{0:t}(\boldsymbol{x}_t) - \sum_{t=0}^{T-1} h_{0:t}(\boldsymbol{x}_{t+1}) - \left( h_{0:T}(\boldsymbol{u}_T) + \tilde{f}_{T+1}(\boldsymbol{u}_T) + \sum_{t=1}^{T-1} h_{0:t}(\boldsymbol{u}_t) - \sum_{t=0}^{T-1} h_{0:t}(\boldsymbol{u}_{t+1}) \right)$$

$$\leq \sum_{t=1}^{T} h_{0:t}(\boldsymbol{x}_t) - \sum_{t=1}^{T-1} h_{0:t}(\boldsymbol{x}_{t+1}) - \left( h_{0:T}(\boldsymbol{x}_{T+1}) + \tilde{f}_{T+1}(\boldsymbol{x}_{T+1}) + \sum_{t=1}^{T-1} h_{0:t}(\boldsymbol{u}_t) - \sum_{t=1}^{T-1} h_{0:t}(\boldsymbol{u}_{t+1}) \right)$$

$$= \sum_{t=1}^{T} h_{0:t}(\boldsymbol{x}_t) - \sum_{t=1}^{T} h_{0:t}(\boldsymbol{x}_{t+1}) - \left( \sum_{t=1}^{T-1} h_{0:t}(\boldsymbol{u}_t) - \sum_{t=1}^{T-1} h_{0:t}(\boldsymbol{u}_{t+1}) \right) - \tilde{f}_{T+1}(\boldsymbol{x}_{T+1}).$$

The inequality holds because of the update rule for each $\boldsymbol{x}_{t+1}$ for any $t$. I.e.,

$$h_{0:T}(\boldsymbol{x}_{T+1}) + \tilde{f}_{T+1}(\boldsymbol{x}_{T+1}) \leq h_{0:T}(\boldsymbol{u}_T) + \tilde{f}_{T+1}(\boldsymbol{u}_T).$$

Also, $h_0(\boldsymbol{u}_{t+1}) = 0, \forall t \leq T-1$.

Writing $h_t$ of the LHS explicitly we get

$$\sum_{t=1}^{T} (\langle \boldsymbol{p}_t, \boldsymbol{x}_t \rangle + r_t(\boldsymbol{x}_t) - \langle \boldsymbol{p}_t, \boldsymbol{u}_t \rangle - r_t(\boldsymbol{u}_t)) - \tilde{f}_{T+1}(\boldsymbol{u}_T)$$

$$\leq \sum_{t=1}^{T} (h_{0:t}(\boldsymbol{x}_t) - h_{0:t}(\boldsymbol{x}_{t+1})) + \sum_{t=1}^{T-1} (h_{0:t}(\boldsymbol{u}_{t+1}) - h_{0:t}(\boldsymbol{u}_t)) - \tilde{f}_{T+1}(\boldsymbol{x}_{T+1}).$$

Since $\tilde{f}_{T+1}(\cdot)$ does not affect the algorithm, we can set it to $0$. Rearranging:

$$\sum_{t=1}^{T} \langle \boldsymbol{p}_t, \boldsymbol{x}_t \rangle - \langle \boldsymbol{p}_t, \boldsymbol{u}_t \rangle \leq \sum_{t=1}^{T} (h_{0:t}(\boldsymbol{x}_t) - h_{0:t}(\boldsymbol{x}_{t+1}) - r_t(\boldsymbol{x}_t) + r_t(\boldsymbol{u}_t)) + \sum_{t=1}^{T-1} (h_{0:t}(\boldsymbol{u}_{t+1}) - h_{0:t}(\boldsymbol{u}_t)) .$$

Noting that by (20), the LHS upper-bounds the regret, we get the result. $\qquad \square$

## B.2. Bounding the First Part (I):

We bound **(I)** : $h_{0:t}(\boldsymbol{x}_t) - h_{0:t}(\boldsymbol{x}_{t+1}) - r_t(\boldsymbol{x}_t)$ in two ways, which results in the $\min(\cdot, \cdot)$ term. We present in this subsection Lemmas B.2 and 4.2, corresponding to each argument of the $\min(\cdot, \cdot)$.

First, we begin with Lemma B.1, which provides an additional characterization of the iterate $\boldsymbol{x}_t$ as the minimizer not only of the original update rule, but also of a related linearized expression.

**Lemma B.1.** *For any $\boldsymbol{x}_t \in \mathcal{X}$, $\boldsymbol{x}_t = \arg\min_{\boldsymbol{x}} h_{0:t-1}(\boldsymbol{x}) + \tilde{f}_t(\boldsymbol{x}) \implies \boldsymbol{x}_t = \arg\min_{\boldsymbol{x}} h_{0:t-1}(\boldsymbol{x}) + \langle \tilde{\boldsymbol{g}}_t, \boldsymbol{x} \rangle + \langle \boldsymbol{g}_t^I, \boldsymbol{x} \rangle$, where $\tilde{\boldsymbol{g}}_t \in \partial \tilde{f}_t(\boldsymbol{x}_t)$ and $\boldsymbol{g}_t^I$ is selected according to (19).*

*Proof.* We are given that $\boldsymbol{x}_t = \arg\min_{\boldsymbol{x}} h_{0:t-1}(\boldsymbol{x}) + \tilde{f}_t(\boldsymbol{x})$. Thus, from the optimality condition for constrained problems (e.g., (Beck, 2017, Thm. 3.67)), the negative (sub)gradient must belong to the normal cone at $\boldsymbol{x}_t$:

$$-\nabla h_{1:t-1}(\boldsymbol{x}_t) - \tilde{\boldsymbol{g}}_t \in \mathcal{N}_{\mathcal{X}}(\boldsymbol{x}_t) \quad \text{(By definition of } \tilde{\boldsymbol{g}}_t\text{).} \tag{21}$$

Next, we examine the optimality condition for $h_{0:t-1}(\boldsymbol{x}) + \langle \tilde{\boldsymbol{g}}_t, \boldsymbol{x} \rangle + \langle \boldsymbol{g}_t^I, \boldsymbol{x} \rangle$ and show that $\boldsymbol{x}_t$ satisfies it. For $\boldsymbol{y}$ to be minimizer of $h_{0:t-1}(\boldsymbol{x}) + \langle \tilde{\boldsymbol{g}}_t, \boldsymbol{x} \rangle + \langle \boldsymbol{g}_t^I, \boldsymbol{x} \rangle$, it must satisfy:

$$-\nabla h_{1:t-1}(\boldsymbol{y}) - \tilde{\boldsymbol{g}}_t - \boldsymbol{g}_t^I \in \mathcal{N}_{\mathcal{X}}(\boldsymbol{y}).$$

Substituting $\boldsymbol{x}_t$ in the above we get

$$-\nabla h_{1:t-1}(\boldsymbol{x}_t) - \tilde{\boldsymbol{g}}_t - \boldsymbol{g}_t^I \in \mathcal{N}_{\mathcal{X}}(\boldsymbol{x}_t). \tag{22}$$

Now, note that according to the linearization choices in (19), we have that either $(i)$: $\boldsymbol{g}_t^I = 0$, or $(ii)$: $\boldsymbol{g}_t^I = -(\boldsymbol{p}_{1:t-1} + \tilde{\boldsymbol{g}}_t + \sigma_{1:t-1}\boldsymbol{x}_t)$.

In case $(i)$, (22) reduces to the given (21), meaning that $\boldsymbol{x}_t$ satisfies (22).

In case $(ii)$, Note that $-(\boldsymbol{p}_{1:t-1} + \tilde{\boldsymbol{g}}_t + \sigma_{1:t-1}\boldsymbol{x}_t) = -\nabla h_{1:t-1}(\boldsymbol{x}_t) - \tilde{\boldsymbol{g}}_t$, and hence (22) reduces to $0 \in \mathcal{N}_{\mathcal{X}}(\boldsymbol{x}_t)$, which is always true. Thus, $\boldsymbol{x}_t$ satisfies (22) in this case too.

It follows that $\boldsymbol{x}_t$ satisfies the optimality condition for $h_{0:t-1}(\boldsymbol{x}) + \langle \tilde{\boldsymbol{g}}_t, \boldsymbol{x} \rangle + \langle \boldsymbol{g}_t^I, \boldsymbol{x} \rangle$ and hence is a minimizer[10] thereof. $\square$

**Lemma B.2.** *Let each function $h_{0:t}(\cdot)$ be 1-strongly convex with respect to a norm $\| \cdot \|_t$ defined as in Lemma 4.1. Let $\boldsymbol{x}_t = \operatorname{argmin}_{\boldsymbol{x}} h_{0:t-1}(\boldsymbol{x}) + \hat{f}_t(\boldsymbol{x})$. Then, we have the inequality*

$$h_{0:t}(\boldsymbol{x}_t) - h_{0:t}(\boldsymbol{x}_{t+1}) - r_t(\boldsymbol{x}_t) \leq 2R\epsilon_t.$$

*Proof.*

$$\begin{aligned}
h_{0:t}(\boldsymbol{x}_t) - h_{0:t}(\boldsymbol{x}_{t+1}) - r_t(\boldsymbol{x}_t) &= h_{0:t-1}(\boldsymbol{x}_t) + \langle \boldsymbol{p}_t, \boldsymbol{x}_t \rangle - h_{0:t-1}(\boldsymbol{x}_{t+1}) - \langle \boldsymbol{p}_t, \boldsymbol{x}_{t+1} \rangle - r_t(\boldsymbol{x}_{t+1}) \\
&\overset{(a)}{\leq} h_{0:t-1}(\boldsymbol{x}_t) + \langle \boldsymbol{p}_t, \boldsymbol{x}_t \rangle - h_{0:t-1}(\boldsymbol{x}_{t+1}) - \langle \boldsymbol{p}_t, \boldsymbol{x}_{t+1} \rangle \\
&\overset{(b)}{\leq} h_{0:t-1}(\boldsymbol{x}_t) + \langle \boldsymbol{p}_t, \boldsymbol{x}_t \rangle - h_{0:t-1}(\boldsymbol{y}_t) - \langle \boldsymbol{p}_t, \boldsymbol{y}_t \rangle, \tag{23}
\end{aligned}$$

where $(a)$ follows by dropping the negative $-r_t(\boldsymbol{x}_{t+1})$ term and $(b)$ by defining $\boldsymbol{y}_t \doteq \arg\min_{\boldsymbol{x}} h_{0:t-1}(\boldsymbol{x}) + \langle \boldsymbol{p}_t, \boldsymbol{x} \rangle$. Then,

$$\begin{aligned}
&h_{0:t-1}(\boldsymbol{x}_t) + \langle \boldsymbol{p}_t, \boldsymbol{x}_t \rangle - h_{0:t-1}(\boldsymbol{y}_t) - \langle \boldsymbol{p}_t, \boldsymbol{y}_t \rangle \\
&= h_{0:t-1}(\boldsymbol{x}_t) + \langle \tilde{\boldsymbol{g}}_t + \boldsymbol{g}_t^I, \boldsymbol{x}_t \rangle + \langle \boldsymbol{p}_t, \boldsymbol{x}_t \rangle - h_{0:t-1}(\boldsymbol{y}_t) - \langle \boldsymbol{p}_t, \boldsymbol{y}_t \rangle - \langle \tilde{\boldsymbol{g}}_t + \boldsymbol{g}_t^I, \boldsymbol{x}_t \rangle \\
&\overset{(c)}{\leq} h_{0:t-1}(\boldsymbol{y}_t) + \langle \tilde{\boldsymbol{g}}_t + \boldsymbol{g}_t^I, \boldsymbol{y}_t \rangle + \langle \boldsymbol{p}_t, \boldsymbol{x}_t \rangle - h_{0:t-1}(\boldsymbol{y}_t) - \langle \boldsymbol{p}_t, \boldsymbol{y}_t \rangle - \langle \tilde{\boldsymbol{g}}_t + \boldsymbol{g}_t^I, \boldsymbol{x}_t \rangle \\
&= \langle \tilde{\boldsymbol{g}}_t + \boldsymbol{g}_t^I, \boldsymbol{y}_t - \boldsymbol{x}_t \rangle + \langle \boldsymbol{p}_t, \boldsymbol{x}_t - \boldsymbol{y}_t \rangle = \langle \boldsymbol{p}_t, \boldsymbol{x}_t - \boldsymbol{y}_t \rangle - \langle \tilde{\boldsymbol{g}}_t + \boldsymbol{g}_t^I, \boldsymbol{x}_t - \boldsymbol{y}_t \rangle \\
&= \langle \boldsymbol{p}_t - \tilde{\boldsymbol{g}}_t - \boldsymbol{g}_t^I, \boldsymbol{x}_t - \boldsymbol{y}_t \rangle = \langle \boldsymbol{g}_t - \tilde{\boldsymbol{g}}_t, \boldsymbol{x}_t - \boldsymbol{y}_t \rangle \leq 2R\|\boldsymbol{g}_t - \tilde{\boldsymbol{g}}_t\| = 2R\epsilon_t,
\end{aligned}$$

where we added & subtracted $\langle \tilde{\boldsymbol{g}}_t + \boldsymbol{g}_t^I, \boldsymbol{x}_t \rangle$ in the first equality, and $(c)$ holds because $\boldsymbol{x}_t$ is the minimizer of $h_{0:t-1}(\boldsymbol{x}) + \langle \tilde{\boldsymbol{g}}_t + \boldsymbol{g}_t^I, \boldsymbol{x} \rangle$ (shown by Lemma B.1). Overall, we obtain

$$h_{0:t}(\boldsymbol{x}_t) - h_{0:t}(\boldsymbol{x}_{t+1}) - r_t(\boldsymbol{x}_t) \leq 2R\epsilon_t. \tag{24}$$

$\square$

To produce the other argument in the $\min(\cdot, \cdot)$, we use the following lemma.

**Lemma 4.2.** *Let each function $h_{0:t}(\cdot)$ be 1-strongly convex with respect to a norm $\| \cdot \|_t$ defined as in Lemma 4.1. Let $\boldsymbol{x}_t = \operatorname{argmin}_{\boldsymbol{x}} h_{0:t-1}(\boldsymbol{x}) + \hat{f}_t(\boldsymbol{x})$. Then, we have the inequality*

$$h_{0:t}(\boldsymbol{x}_t) - h_{0:t}(\boldsymbol{x}_{t+1}) - r_t(\boldsymbol{x}_t) \leq \frac{1}{2}\|\boldsymbol{g}_t - \tilde{\boldsymbol{g}}_t\|_{t-1,*}^2 .$$

*Proof.*

$$h_{0:t}(\boldsymbol{x}_t) - h_{0:t}(\boldsymbol{x}_{t+1}) - r_t(\boldsymbol{x}_t)$$

---

[10]Since $h_{0:t}(\cdot)$ is strongly convex, the minimizer of the two expressions in the lemma statement always exist and unique. In the case $\sigma_{1:t} = 0$, we abuse notation by writing " $= \operatorname{argmin}$ " instead of " $\in \operatorname{argmin}$ ". This is not problematic because we do not require the equivalence of the minimizers.

$$= h_{0:t-1}(\boldsymbol{x}_t) + \langle \boldsymbol{p}_t, \boldsymbol{x} \rangle - h_{0:t-1}(\boldsymbol{x}_{t+1}) - \langle \boldsymbol{p}_t, \boldsymbol{x}_{t+1} \rangle - r_t(\boldsymbol{x}_{t+1})$$
$$\leq h_{0:t-1}(\boldsymbol{x}_t) + \langle \boldsymbol{p}_t, \boldsymbol{x}_t \rangle - h_{0:t-1}(\boldsymbol{x}_{t+1}) - \langle \boldsymbol{p}_t, \boldsymbol{x}_{t+1} \rangle$$
$$\leq h_{0:t-1}(\boldsymbol{x}_t) + \langle \boldsymbol{p}_t, \boldsymbol{x}_t \rangle - h_{0:t-1}(\boldsymbol{y}_t) - \langle \boldsymbol{p}_t, \boldsymbol{y}_t \rangle, \quad \boldsymbol{y}_t \doteq \operatorname*{argmin}_{\boldsymbol{x}} h_{0:t-1}(\boldsymbol{x}) + \langle \boldsymbol{p}_t, \boldsymbol{x} \rangle$$

Where again the inequality follows by dropping the non-positive $-r_t(\cdot)$. Next, we invoke Lemma D.1 with

$$\phi_1(\boldsymbol{x}) \doteq h_{0:t-1}(\boldsymbol{x}) + \langle \tilde{\boldsymbol{g}}_t, \boldsymbol{x} \rangle + \langle \boldsymbol{g}_t^I, \boldsymbol{x} \rangle,$$
$$\phi_2(\boldsymbol{x}) \doteq h_{0:t-1}(\boldsymbol{x}) + \langle \boldsymbol{p}_t, \boldsymbol{x} \rangle = h_{0:t-1}(\boldsymbol{x}) + \langle \boldsymbol{g}_t, \boldsymbol{x} \rangle + \langle \boldsymbol{g}_t^I, \boldsymbol{x} \rangle = \phi_1(\boldsymbol{x}) + \underbrace{\langle \boldsymbol{g}_t - \tilde{\boldsymbol{g}}_t, \boldsymbol{x} \rangle}_{\psi(\boldsymbol{x})}.$$

Under these definitions, we indeed have that

$$\boldsymbol{x}_t = \operatorname*{argmin}_{\boldsymbol{x}} h_{0:t-1}(\boldsymbol{x}) + \tilde{f}_t(\boldsymbol{x})$$
$$\overset{(\text{Lem. B.1})}{=} \operatorname*{argmin}_{\boldsymbol{x}} h_{0:t-1}(\boldsymbol{x}) + \langle \tilde{\boldsymbol{g}}_t, \boldsymbol{x} \rangle + \langle \boldsymbol{g}_t^I, \boldsymbol{x} \rangle = \arg\min_{\boldsymbol{x}} \phi_1(\boldsymbol{x}) = \boldsymbol{y}_1.$$

and thus selecting $\boldsymbol{y}'$ of Lemma D.1 as $\boldsymbol{y}_t$, the result of the same (Lemma D.1) gives:

$$h_{0:t-1}(\boldsymbol{x}_t) + \langle \boldsymbol{p}_t, \boldsymbol{x}_t \rangle - h_{0:t-1}(\boldsymbol{y}_t) - \langle \boldsymbol{p}_t, \boldsymbol{y}_t \rangle \leq \frac{1}{2} \|\boldsymbol{g}_t - \tilde{\boldsymbol{g}}_t\|_{t-1,*}^2 , \tag{25}$$

noticing that $\partial \psi(\boldsymbol{x}_t) = \boldsymbol{g}_t - \tilde{\boldsymbol{g}}_t$. $\qquad\square$

From (25) and (24), we have that

$$(\mathbf{I}) \leq \min\left( \frac{1}{2} \|\boldsymbol{g}_t - \tilde{\boldsymbol{g}}_t\|_{t-1,*}^2, 2R\epsilon_t \right) \tag{26}$$

### B.3. Bounding the Second Part (II):

The second part does not have the structure exploited in the first one ($\boldsymbol{u}_t$ and $\boldsymbol{u}_{t+1}$ are arbitrary). Hence, we must resort to the strong convexity property to obtain

$$h_{0:t}(\boldsymbol{u}_{t+1}) - h_{0:t}(\boldsymbol{u}_t) \leq \langle \boldsymbol{q}_t, \boldsymbol{u}_{t+1} - \boldsymbol{u}_t \rangle - \frac{\sigma_{1:t}}{2} \|\boldsymbol{u}_{t+1} - \boldsymbol{u}_t\|^2$$
$$\leq \|\boldsymbol{q}_t\| \|\boldsymbol{u}_{t+1} - \boldsymbol{u}_t\| - \frac{\sigma_{1:t}}{2} \|\boldsymbol{u}_{t+1} - \boldsymbol{u}_t\|^2,$$

where

$$\boldsymbol{q}_t \in \partial h_{0:t}(\boldsymbol{u}_{t+1}).$$

Nonetheless, we will still exploit problem properties to bound $\|\boldsymbol{q}_t\|$:

$$\partial h_{0:t}(\boldsymbol{x}) = \boldsymbol{p}_{1:t} + \sum_{\tau=1}^{t} \sigma_\tau \boldsymbol{x} + \mathcal{N}_{\mathcal{X}}(\boldsymbol{x}),$$

which gives

$$\boldsymbol{q}_t = \boldsymbol{p}_{1:t} + \sum_{\tau=1}^{t} \sigma_\tau \boldsymbol{u}_{t+1},$$

where we chose the 0 vector from $\mathcal{N}_{\mathcal{X}}(\boldsymbol{u}_{t+1})$. Hence, the length of $\boldsymbol{q}_t$ satisfies:

$$\|\boldsymbol{q}_t\| \leq \|\boldsymbol{p}_{1:t}\| + \sum_{\tau=1}^{t} \sigma_\tau \|\boldsymbol{u}_{t+1}\| = \|\boldsymbol{p}_{1:t}\| + \sigma_{1:t} \|\boldsymbol{u}_{t+1}\| \leq \|\boldsymbol{p}_{1:t}\| + R\sigma_{1:t}$$

Overall

$$h_{0:t}(\boldsymbol{u}_{t+1}) - h_{0:t}(\boldsymbol{u}_t) \leq (\|\boldsymbol{p}_{1:t}\| + R\sigma_{1:t}) \|\boldsymbol{u}_{t+1} - \boldsymbol{u}_t\| - \frac{\sigma_{1:t}}{2} \|\boldsymbol{u}_{t+1} - \boldsymbol{u}_t\|^2. \tag{27}$$

From (27) it follows that

$$\mathbf{(II)} \leq (\|\boldsymbol{p}_{1:t}\| + R\sigma_{1:t}) \|\boldsymbol{u}_{t+1} - \boldsymbol{u}_t\| - \frac{\sigma_{1:t}}{2} \|\boldsymbol{u}_{t+1} - \boldsymbol{u}_t\|^2 \tag{28}$$

$$\leq (\|\boldsymbol{p}_{1:t}\| + R\sigma_{1:t}) \|\boldsymbol{u}_{t+1} - \boldsymbol{u}_t\| \tag{29}$$

Note that the negative term in (28) admits a tight lower bound of $0$ and will therefore be omitted. Next, we derive a bound on the state vector's length $\|\boldsymbol{p}_{1:t}\|$.

**Lemma 4.3.** *(Optimistically Bounded State) Let $\{\boldsymbol{p}_t\}_{t=1}^T$ be a sequence of vectors such that each $\boldsymbol{p}_t = \boldsymbol{g}_t + \boldsymbol{g}_t^I$, where $\boldsymbol{g}_t \in \partial f_t(\boldsymbol{x}_t)$, and $\boldsymbol{g}_t^I \in \partial I_\mathcal{X}(\boldsymbol{x}_t)$ is chosen according to the construction in (5). That is, $\boldsymbol{p}_t$ corresponds to the linearization of the composite function $f_t(\cdot) + I_\mathcal{X}(\cdot)$ around $\boldsymbol{x}_t$. Then, for any $t$, the following holds:*

$$\|\boldsymbol{p}_{1:t}\| \leq R\sigma_{1:t-1} + \epsilon_t.$$

*Proof.* The proof of this lemma was stated in the paper. $\qquad\square$

## C. Missing Proofs for Section 3

Recall the problem parameters described by the following settings:

**Settings 1:** Let $\mathcal{X} \subset \mathbb{R}^d$ be a compact, convex set such that $\|\boldsymbol{x}\| \leq R$ for all $\boldsymbol{x} \in \mathcal{X}$. Let $\{f_t(\cdot), \tilde{f}_t(\cdot)\}_{t=1}^T$ be any sequence of $L$-Lipschitz convex functions, and define the following quantities.

$$\text{The cumulative-squared prediction error: } E_T \doteq \sum_{t=1}^T \epsilon_t^2, \quad \epsilon_t \doteq \|\boldsymbol{g}_t - \tilde{\boldsymbol{g}}_t\|.$$

$$\text{The path length: } \qquad P_T \doteq \sum_{t=1}^{T-1} \|\boldsymbol{u}_{t+1} - \boldsymbol{u}_t\|$$

$$\text{The prediction-weighted path length: } \quad H_T \doteq \sum_{t=1}^{T-1} \epsilon_t \|\boldsymbol{u}_{t+1} - \boldsymbol{u}_t\|$$

### C.1. Proof of Theorem 3.1

Consider the following regularization strategy

$$\sigma = \frac{1}{4R}, \text{ and } \sigma_1 = \sigma\epsilon_1, \ \sigma_t = \sigma\left(\sqrt{E_t} - \sqrt{E_{t-1}}\right), \forall t \geq 2. \tag{30}$$

**Theorem 3.1.** *Under Settings 1, Alg. 1 run with the regularization strategy in (30) produces points $\{\boldsymbol{x}_t\}_{t=1}^T$ such that, for any $T$, the dynamic regret $\mathcal{R}_T$ satisfies:*

$$\mathcal{R}_T \leq \left(5.8R + (1/2)P_T\right)\sqrt{E_T} + H_T = \mathcal{O}\left((1 + P_T)\sqrt{E_T}\right).$$

*Proof.* We begin by substituting the bounds of **(I)** and **(II)**, from (26) and (29), respectively, back in the result of Lemma 4.1:

$$\mathcal{R}_T \leq \sum_{t=1}^T \left(\min\left(\frac{1}{2}\|\boldsymbol{g}_t - \tilde{\boldsymbol{g}}_t\|_{t-1,*}^2, 2R\epsilon_t\right) + r_t(\boldsymbol{u}_t)\right) + \sum_{t=1}^{T-1} \left((R\sigma_{1:t} + \|\boldsymbol{p}_{1:t}\|) \|\boldsymbol{u}_{t+1} - \boldsymbol{u}_t\|\right)$$

$$\overset{(a)}{\leq} \sum_{t=1}^T \left(\min\left(\frac{1}{2}\|\boldsymbol{g}_t - \tilde{\boldsymbol{g}}_t\|_{t-1,*}^2, 2R\epsilon_t\right) + r_t(\boldsymbol{u}_t)\right) + \sum_{t=1}^{T-1} \left((2R\sigma_{1:t} + \epsilon_t) \|\boldsymbol{u}_{t+1} - \boldsymbol{u}_t\|\right)$$

$$\overset{(b)}{\leq} \sum_{t=1}^{T} \min\left(\frac{1}{2}\|g_t - \tilde{g}_t\|_{t-1,*}^2, 2R\epsilon_t\right) + \frac{R^2}{2}\sigma_{1:T} + \sum_{t=1}^{T-1}\left((2R\sigma_{1:t} + \epsilon_t)\|u_{t+1} - u_t\|\right)$$

$$= \sum_{t=1}^{T} \min\left(\frac{\epsilon_t^2}{2\sigma_{0:t-1}}, 2R\epsilon_t\right) + \frac{R^2}{2}\sigma_{1:T} + \sum_{t=1}^{T-1}\left((2R\sigma_{1:t} + \epsilon_t)\|u_{t+1} - u_t\|\right)$$

$$= \sum_{t=1}^{T} \min\left(\frac{\epsilon_t^2}{2\sigma\sqrt{E_{t-1}}}, 2R\epsilon_t\right) + \frac{R^2}{2}\sigma\sqrt{E_T} + \sum_{t=1}^{T-1}\left(\left(2R\sigma\sqrt{E_t} + \epsilon_t\right)\|u_{t+1} - u_t\|\right)$$

$$\overset{(c)}{\leq} 4\sqrt{2}R\sqrt{E_T} + \frac{R}{8}\sqrt{E_T} + \sum_{t=1}^{T-1}\frac{1}{2}\sqrt{E_t}\|u_{t+1} - u_t\| + H_T$$

$$\overset{(d)}{\leq} 5.8R\sqrt{E_T} + \frac{1}{2}\sqrt{E_{T-1}}\sum_{t=1}^{T-1}\|u_{t+1} - u_t\| + H_T$$

$$= 5.8R\sqrt{E_T} + \frac{1}{2}\sqrt{E_T}P_T + H_T$$

where $(a)$ follows by Lemma 4.3 which bounds $\|p_{1:t}\|$, and by the fact that $\sigma_{1:t-1} \leq \sigma_{1:t}$, $(b)$ by bounding each $\|u_t\|$ in $r_t(u_t)$ by $R$, $(c)$ by Lemma D.3, which is a common tool to bound the sum of a decreasing function (on $E_{t-1}$), with the choice of $\sigma = \frac{1}{4R}$, and $(d)$ by the fact that $\sqrt{E_t}$ are non-decreasing. $\square$

*Remark* C.1. On the growth rate of the hybrid term $H_T$.

Note that by Cauchy-Schwarz, we have that

$$H_T = \sum_{t=1}^{T-1}\epsilon_t\|u_{t+1} - u_t\| \leq \sqrt{\sum_{t=1}^{T-1}\epsilon_t^2}\sqrt{\sum_{t=1}^{T-1}\|u_{t+1} - u_t\|^2} \leq \sqrt{2R}\sqrt{\sum_{t=1}^{T-1}\epsilon_t^2}\sqrt{\sum_{t=1}^{T-1}\|u_{t+1} - u_t\|} = \sqrt{2R}\sqrt{E_{T-1}}\sqrt{P_T}.$$

Hence, $H_T$ is also $\mathcal{O}\left(\sqrt{P_T E_T}\right)$.

### C.2. Proof of Theorem 3.2

Consider the following regularization strategy:

$$\sigma = \frac{1}{2\sqrt{2RP_T'}}, \quad \text{where } P_T' \text{ is the augmented path length:} \quad P_T' \doteq 2R + P_T.$$

$$\sigma_1 = \sigma\epsilon_1, \ \sigma_t = \sigma\left(\sqrt{E_t} - \sqrt{E_{t-1}}\right), \forall t \geq 2. \tag{31}$$

**Theorem 3.2.** *Under Settings 1, Alg. 1 run with the regularization strategy in* (31) *produces points* $\{x_t\}_{t=1}^{T}$ *such that, for any* $T$, *the dynamic regret* $\mathcal{R}_T$ *satisfies:*

$$\mathcal{R}_T \leq \left(4\sqrt{2R^2 + P_T} + \frac{R}{8} + \sqrt{\frac{RP_T}{2}}\right)\sqrt{E_T} + H_T = \mathcal{O}\left((1 + \sqrt{P_T})\sqrt{E_T}\right).$$

*Proof.* Similarly to Theorem 1, we begin by substituting the bounds of (**I**) and (**II**), from (26) and (29), respectively, back in the result of Lemma 4.1:

$$\mathcal{R}_T \leq \sum_{t=1}^{T}\left(\min\left(\frac{1}{2}\|g_t - \tilde{g}_t\|_{t-1,*}^2, 2R\epsilon_t\right) + r_t(u_t)\right) + \sum_{t=1}^{T-1}\left((R\sigma_{1:t} + \|p_{1:t}\|)\|u_{t+1} - u_t\|\right)$$

$$\overset{(a)}{\leq} \sum_{t=1}^{T}\left(\min\left(\frac{1}{2}\|g_t - \tilde{g}_t\|_{t-1,*}^2, 2R\epsilon_t\right) + r_t(u_t)\right) + \sum_{t=1}^{T-1}\left((2R\sigma_{1:t} + \epsilon_t)\|u_{t+1} - u_t\|\right)$$

$$\overset{(b)}{\leq} \sum_{t=1}^{T}\min\left(\frac{1}{2}\|g_t - \tilde{g}_t\|_{t-1,*}^2, 2R\epsilon_t\right) + \frac{R^2}{2}\sigma_{1:T} + \sum_{t=1}^{T-1}\left((2R\sigma_{1:t} + \epsilon_t)\|u_{t+1} - u_t\|\right)$$

$$= \sum_{t=1}^{T} \min\left(\frac{\epsilon_t^2}{2\sigma_{0:t-1}}, 2R\epsilon_t\right) + \frac{R^2}{2}\sigma_{1:T} + \sum_{t=1}^{T-1}\left((2R\sigma_{1:t} + \epsilon_t)\|\boldsymbol{u}_{t+1} - \boldsymbol{u}_t\|\right)$$

$$= \sum_{t=1}^{T} \min\left(\frac{\epsilon_t^2}{2\sigma\sqrt{E_{t-1}}}, 2R\epsilon_t\right) + \frac{R^2}{2}\sigma\sqrt{E_T} + \sum_{t=1}^{T-1}\left(\left(2R\sigma\sqrt{E_t} + \epsilon_t\right)\|\boldsymbol{u}_{t+1} - \boldsymbol{u}_t\|\right)$$

$$\overset{(c)}{\leq} 4\sqrt{R}\sqrt{P_T' E_T} + \frac{R^2}{4\sqrt{2RP_T'}}\sqrt{E_T} + \sum_{t=1}^{T-1}\left(\left(\frac{R}{\sqrt{2RP_T'}}\sqrt{E_t} + \epsilon_t\right)\|\boldsymbol{u}_{t+1} - \boldsymbol{u}_t\|\right)$$

$$\overset{(d)}{\leq} 4\sqrt{R}\sqrt{P_T' E_T} + \frac{R}{8}\sqrt{E_T} + \sqrt{\frac{R}{2}}\sum_{t=1}^{T-1}\sqrt{E_t}\frac{\|\boldsymbol{u}_{t+1} - \boldsymbol{u}_t\|}{\sqrt{P_T}} + H_T$$

$$\overset{(e)}{\leq} 4\sqrt{R}\sqrt{P_T' E_T} + \frac{R}{8}\sqrt{E_T} + \sqrt{\frac{R}{2}}\sqrt{E_{T-1}}\frac{P_T}{\sqrt{P_T}} + H_T$$

$$\leq 4\sqrt{R}\sqrt{P_T' E_T} + \frac{R}{8}\sqrt{E_T} + \sqrt{\frac{R}{2}}\sqrt{E_T P_T} + H_T$$

$$\leq \left(4\sqrt{RP_T'} + \frac{R}{8} + \sqrt{\frac{R}{2}}\sqrt{P_T}\right)\sqrt{E_T} + H_T$$

$$\leq \left(4\sqrt{R(2R + P_T)} + \frac{R}{8} + \sqrt{\frac{R}{2}}\sqrt{P_T}\right)\sqrt{E_T} + H_T$$

$$= \mathcal{O}\left((1 + \sqrt{P_T})\sqrt{E_T}\right),$$

where $(a)$ follows by Lemma 4.3 which bounds $\|\boldsymbol{p}_{1:t}\|$, and by $\sigma_{1:t-1} \leq \sigma_{1:t}$, $(b)$ by bounding each $\|\boldsymbol{u}_t\|$ in $r_t(\boldsymbol{u}_t), t \leq T$ by $R$, $(c)$ by lemma D.3, with the choice of $\sigma = \frac{1}{2\sqrt{2RP_T'}}$, (note that this choice still satisfies the Lemma's condition since $P_T' \geq 2R$),$(d)$ also used that used $2R \leq P_T'$, $(e)$ used that $E_t$ is non-decreasing, and finally the $\mathcal{O}(\cdot)$ expression follows from Remark C.1. $\qquad\square$

*Remark* C.2. On guaranteeing $\sqrt{P_T E_T}, \forall u_t$.

To obtain a minimax bound that holds uniformly over all comparator sequences (i.e., without assuming prior knowledge of their path length, and without assuming that they are observable online), one can instantiate $\Theta(\log T)$ sub-learners, each with a halving $\sigma$ starting from $1/\sqrt{T}$. Then, using the meta-learner of (Zhao et al., 2020), the minimax bound can be recovered. The gist of this approach is that eventually $\exists$ an expert $i$ such that $\forall P_T, \sigma^{(i)} \geq 1/\sqrt{P_T} \geq 1/2\sigma^{(i)}$.

### C.3. Proof of Theorem 3.3

Define the augmented seen path length at $t$ as:

$$P_t' \doteq 2R + P_t = 2R + \sum_{\tau=1}^{t-1}\|\boldsymbol{u}_{\tau+1} - \boldsymbol{u}_\tau\|.$$

and consider the following regularization strategy

$$\sigma = \frac{1}{2\sqrt{2R}}, \quad \sigma_1 = \frac{\sigma\epsilon_1}{\sqrt{P_1'}}, \quad \sigma_t = \sigma\max\left(0, \sqrt{\frac{E_t}{P_t'}} - \sqrt{\frac{E_{t-1}}{P_{t-1}'}}\right), \forall t \geq 2. \tag{32}$$

**Theorem 3.3.** *Under Settings 1, Alg. 1 run with the regularization strategy in* (32) *produces points* $\{\boldsymbol{x}_t\}_{t=1}^{T}$ *such that, for any $T$, the dynamic regret $\mathcal{R}_T$ satisfies:*

$$\mathcal{R}_T \leq 5.5\sqrt{R}\sqrt{E_T P_T'} + H_T + \sqrt{R/2}\,A_T = \mathcal{O}\left((1 + \sqrt{P_T})\sqrt{E_T} + A_T\right) \quad where$$

$$A_T \doteq \sum_{t=1}^{T}\sum_{\tau \in [t]^+}\left(\sqrt{\frac{E_{\tau-1}}{P_{\tau-1}'}} - \sqrt{\frac{E_\tau}{P_\tau'}}\right)\|\boldsymbol{u}_{t+1} - \boldsymbol{u}_t\|, \quad with \quad [t]^+ = \left\{2 \leq \tau \leq t \,\Big|\, \sqrt{\frac{E_{\tau-1}}{P_{\tau-1}'}} - \sqrt{\frac{E_\tau}{P_\tau'}} \geq 0\right\}.$$

Before proceeding to prove the theorem, we will make use of the following two lemmas, which we present independently to streamline the presentation.

**Lemma C.3.** *The squared norm* $\|g_t - \tilde{g}_t\|_{t-1}^2 = \sigma_{0:t-1}\|g_t - \tilde{g}_t\|^2 = \sigma_{0:t-1}\epsilon_t^2$ *can be written as:*

$$\|g_t - \tilde{g}_t\|_{0,*}^2 = 0,$$

$$\|g_t - \tilde{g}_t\|_{t-1,*}^2 \leq \frac{1}{\sigma}\sqrt{\frac{P'_{t-1}}{E_{t-1}}}\,\epsilon_t^2, \qquad \forall t \geq 1$$

*Proof.* Recall first that the dual norm of $\|\cdot\|_{t-1,*}$ is $\frac{1}{\sqrt{\sigma_{0:t-1}}}\|\cdot\|$. The first part is immediate from $\|x\|_{0,*} = \sqrt{\frac{E_0}{P'_0}}\|x\| = 0$, since $E_0 = 0$ and $P'_0 = 2R$. For the second part:

$$\|g_t - \tilde{g}_t\|_{t-1,*}^2 = \frac{\|g_t - \tilde{g}_t\|^2}{\sigma_{1:t-1}} = \frac{\epsilon_t^2}{\frac{\sigma\epsilon_1}{\sqrt{P'_1}} + \sum_{\tau=2}^{t-1}\sigma\max\left(0,\ \sqrt{\frac{E_\tau}{P'_\tau}} - \sqrt{\frac{E_{\tau-1}}{P'_{\tau-1}}}\right)}$$

$$\leq \frac{\epsilon_t^2}{\frac{\sigma\epsilon_1}{\sqrt{P'_1}} + \sum_{\tau=2}^{t-1}\sigma\left(\sqrt{\frac{E_\tau}{P'_\tau}} - \sqrt{\frac{E_{\tau-1}}{P'_{\tau-1}}}\right)} = \frac{\epsilon_t^2}{\sigma\sqrt{\frac{E_{t-1}}{P'_{t-1}}}},$$

where the inequality follows by dropping the $\max$ in the denominator. $\qquad\square$

**Lemma C.4.** *The strong convexity parameter $\sigma_{1:t}$ in* (32) *can be written as the cumulative term* $\sqrt{\frac{E_t}{P'_t}}$ *plus a corrective term:*

$$\sigma_{1:t} = \sigma\left(\sqrt{\frac{E_t}{P'_t}} + \sum_{\tau\in[t]^+}\sqrt{\frac{E_{\tau-1}}{P'_{\tau-1}}} - \sqrt{\frac{E_\tau}{P'_\tau}}\right).$$

*Proof.*

$$\sigma_{1:t} = \frac{\sigma\epsilon_1}{\sqrt{P'_1}} + \sigma\sum_{\tau=2}^{t}\max\left(0,\ \sqrt{\frac{E_\tau}{P'_\tau}} - \sqrt{\frac{E_{\tau-1}}{P'_{\tau-1}}}\right)$$

$$= \sigma\left(\frac{\epsilon_1}{\sqrt{P'_1}} + \sum_{\tau=2}^{t}\sqrt{\frac{E_\tau}{P'_\tau}} - \sqrt{\frac{E_{\tau-1}}{P'_{\tau-1}}}\right) + \sigma\sum_{\tau\in[t]^+}\sqrt{\frac{E_{\tau-1}}{P'_{\tau-1}}} - \sqrt{\frac{E_\tau}{P'_\tau}},$$

where the last equality holds from the definition of $[t]^+$. That is, for the slot $\tau$ when the $\max$ evaluates to 0, we write it as $\left(\sqrt{\frac{E_\tau}{P'_\tau}} - \sqrt{\frac{E_{\tau-1}}{P'_{\tau-1}}}\right) + \left(\sqrt{\frac{E_{\tau-1}}{P'_{\tau-1}}} - \sqrt{\frac{E_\tau}{P'_\tau}}\right)$. Now, by telescoping

$$\sigma_{1:t} = \sigma\left(\frac{\epsilon_1}{\sqrt{P'_1}} + \sqrt{\frac{E_t}{P'_t}} - \sqrt{\frac{E_1}{P'_1}}\right) + \sigma\sum_{\tau\in[t]^+}\sqrt{\frac{E_{\tau-1}}{P'_{\tau-1}}} - \sqrt{\frac{E_\tau}{P'_\tau}}$$

$$= \sigma\sqrt{\frac{E_t}{P'_t}} + \sigma\sum_{\tau\in[t]^+}\sqrt{\frac{E_{\tau-1}}{P'_{\tau-1}}} - \sqrt{\frac{E_\tau}{P'_\tau}}.$$

$\qquad\square$

*Proof of Theorem 3.3.* We begin by substituting the bounds of **(I)** and **(II)**, in (26) and (29), respectively, back in the result of Lemma 4.1:

$$\mathcal{R}_T \leq \sum_{t=1}^{T}\left(\min\left(\frac{1}{2}\|g_t - \tilde{g}_t\|_{t-1,*}^2, 2R\epsilon_t\right) + r_t(u_t)\right) + \sum_{t=1}^{T-1}\left((R\sigma_{1:t} + \|p_{1:t}\|)\|u_{t+1} - u_t\|\right)$$

$$\stackrel{(a)}{\leq} \sum_{t=1}^{T} \left( \min\left( \frac{1}{2}\|\boldsymbol{g}_t - \tilde{\boldsymbol{g}}_t\|^2_{t-1,*}, 2R\epsilon_t \right) + r_t(\boldsymbol{u}_t) \right) + \sum_{t=1}^{T-1} \left( (2R\sigma_{1:t} + \epsilon_t)\|\boldsymbol{u}_{t+1} - \boldsymbol{u}_t\| \right)$$

$$\stackrel{(b)}{\leq} \sum_{t=1}^{T} \min\left( \frac{1}{2}\|\boldsymbol{g}_t - \tilde{\boldsymbol{g}}_t\|^2_{t-1,*}, 2R\epsilon_t \right) + \frac{R^2}{2}\sigma_{1:T} + \sum_{t=1}^{T-1} 2R\sigma_{1:t}\|\boldsymbol{u}_{t+1} - \boldsymbol{u}_t\| + H_T$$

$$\stackrel{(c)}{\leq} \sum_{t=1}^{T} \min\left( \frac{\sqrt{P'_{t-1}}\epsilon_t^2}{2\sigma\sqrt{E_{t-1}}}, 2R\epsilon_t \right) + \sum_{t=1}^{T} 2R\sigma_{1:t}\|\boldsymbol{u}_{t+1} - \boldsymbol{u}_t\| + H_T$$

$$\stackrel{(d)}{\leq} \sum_{t=1}^{T} \min\left( \frac{\sqrt{P'_{t-1}}\epsilon_t^2}{2\sigma\sqrt{E_{t-1}}}, 2R\epsilon_t \right) + 2R\sigma\sum_{t=1}^{T}\sqrt{\frac{E_t}{P'_t}}\|\boldsymbol{u}_{t+1} - \boldsymbol{u}_t\| + H_T$$

$$+ 2R\sigma \underbrace{\sum_{t=1}^{T}\sum_{\tau\in[t]^+}\left( \sqrt{\frac{E_{\tau-1}}{P'_{\tau-1}}} - \sqrt{\frac{E_\tau}{P'_\tau}} \right)\|\boldsymbol{u}_{t+1} - \boldsymbol{u}_t\|}_{\doteq A_T}$$

$$\stackrel{(e)}{\leq} \sum_{t=1}^{T} \sqrt{P'_T}\min\left( \frac{\epsilon_t^2}{2\sigma\sqrt{E_{t-1}}}, \frac{2R}{\sqrt{P'_T}}\epsilon_t \right) + 2R\sigma\sum_{t=1}^{T}\sqrt{\frac{E_t}{P'_t}}\|\boldsymbol{u}_{t+1} - \boldsymbol{u}_t\| + H_T + 2R\sigma A_T$$

$$\stackrel{(f)}{\leq} \sum_{t=1}^{T} \sqrt{P'_T}\min\left( \frac{\epsilon_t^2}{2\sigma\sqrt{E_{t-1}}}, \sqrt{2R}\epsilon_t \right) + 2R\sigma\sqrt{E_T}\sum_{t=1}^{T}\frac{\|\boldsymbol{u}_{t+1} - \boldsymbol{u}_t\|}{\sqrt{P'_t}} + H_T + 2R\sigma A_T$$

$$\stackrel{(g)}{\leq} 4\sqrt{R}\sqrt{P'_T E_T} + \sqrt{\frac{R}{2}}\sqrt{E_T}\sum_{t=1}^{T}\frac{\|\boldsymbol{u}_{t+1} - \boldsymbol{u}_t\|}{\sqrt{2R + \sum_{\tau=1}^{t-1}\|\boldsymbol{u}_{\tau+1} - \boldsymbol{u}_\tau\|}} + H_T + \sqrt{\frac{R}{2}}A_T$$

$$\stackrel{(h)}{\leq} 4\sqrt{R}\sqrt{P'_T E_T} + \sqrt{\frac{R}{2}}\sqrt{E_T}\sum_{t=1}^{T}\frac{\|\boldsymbol{u}_{t+1} - \boldsymbol{u}_t\|}{\sqrt{\sum_{\tau=1}^{t}\|\boldsymbol{u}_{\tau+1} - \boldsymbol{u}_\tau\|}} + H_T + \sqrt{\frac{R}{2}}A_T$$

$$\stackrel{(i)}{\leq} 4\sqrt{R}\sqrt{P'_T E_T} + \sqrt{2R}\sqrt{E_T P'_T} + H_T + \sqrt{\frac{R}{2}}A_T$$

$$= (4 + \sqrt{2})\sqrt{R}\sqrt{P'_T E_T} + H_T + \sqrt{\frac{R}{2}}A_T$$

where $(a)$ follows by Lemma 4.3 which bounds $\|\boldsymbol{p}_{1:t}\|$, and the fact that $\sigma_{1:t-1} \leq \sigma_{1:t}$, $(b)$ by bounding each $\|\boldsymbol{u}_t\|$ in $r_t(\boldsymbol{u}_t), t \leq T$ by $R$, $(c)$ by using Lemma C.3 for the first sum, and by selecting[11] $\boldsymbol{u}_{T+1}$ such that $\|\boldsymbol{u}_{T+1} - \boldsymbol{u}_T\| \geq \frac{R}{4}$, which allows us to append the $\frac{R^2}{2}\sigma_{1:T}$ term to the second sum as the summand with index $T$, $(d)$ by using Lemma C.4 to re-write $\sigma_{1:t}$, $(e)$ since $P'_t$ is non-decreasing, $(f)$ from $P'_t \geq 2R$, $(g)$ by $\sigma = \frac{1}{2\sqrt{2R}}$ and Lemma D.3, $(h)$ by the fact that $2R \geq \|\boldsymbol{u}_{t+1} - \boldsymbol{u}_t\|, \forall t$, and finally $(i)$ by Lemma D.2 with $a_t = \|\boldsymbol{u}_{t+1} - \boldsymbol{u}_t\|$. $\qquad\square$

### C.3.1. BOUNDING THE $A_T$ TERM

In this subsection, we show that the term $A_T$ cannot be worse than the result of Theorem 1 in all cases.

$$A_T \doteq \sum_{t=1}^{T}\sum_{\tau\in[t]^+}\left( \sqrt{\frac{E_{\tau-1}}{P'_{\tau-1}}} - \sqrt{\frac{E_\tau}{P'_\tau}} \right)\|\boldsymbol{u}_{t+1} - \boldsymbol{u}_t\| = \sum_{t=1}^{T}\left[ \|\boldsymbol{u}_{t+1} - \boldsymbol{u}_t\|\sum_{\tau\in[t]^+}\left( \sqrt{\frac{E_{\tau-1}}{P'_{\tau-1}}} - \sqrt{\frac{E_\tau}{P'_\tau}} \right) \right]$$

$$\stackrel{(a)}{\leq} \sum_{t=1}^{T}\left[ \|\boldsymbol{u}_{t+1} - \boldsymbol{u}_t\|\sum_{\tau\in[t]^+}\left( \sqrt{\frac{E_\tau}{P'_{\tau-1}}} - \sqrt{\frac{E_\tau}{P'_\tau}} \right) \right]$$

$$= \sum_{t=1}^{T}\left[ \|\boldsymbol{u}_{t+1} - \boldsymbol{u}_t\|\sum_{\tau\in[t]^+}\sqrt{E_\tau}\left( \sqrt{\frac{1}{P'_{\tau-1}}} - \sqrt{\frac{1}{P'_\tau}} \right) \right]$$

---

[11] Note that $\boldsymbol{u}_{T+1}$ does not affect the algorithm and hence we can set it without loss of generality

$$\overset{(b)}{\leq} \sum_{t=1}^{T} \left[ \|\boldsymbol{u}_{t+1} - \boldsymbol{u}_t\| \sqrt{E_t} \sum_{\tau \in [t]^+} \left( \sqrt{\frac{1}{P'_{\tau-1}}} - \sqrt{\frac{1}{P'_\tau}} \right) \right]$$

$$\overset{(c)}{\leq} \sum_{t=1}^{T} \left[ \|\boldsymbol{u}_{t+1} - \boldsymbol{u}_t\| \sqrt{E_t} \sum_{\tau=2}^{t} \left( \sqrt{\frac{1}{P'_{\tau-1}}} - \sqrt{\frac{1}{P'_\tau}} \right) \right]$$

$$\overset{(d)}{=} \sum_{t=1}^{T} \left[ \|\boldsymbol{u}_{t+1} - \boldsymbol{u}_t\| \sqrt{E_t} \left( \sqrt{\frac{1}{P'_1}} - \sqrt{\frac{1}{P'_t}} \right) \right]$$

$$\overset{(e)}{=} \sum_{t=1}^{T} \left[ \|\boldsymbol{u}_{t+1} - \boldsymbol{u}_t\| \sqrt{E_t} \sqrt{\frac{1}{P'_1}} \right]$$

$$\overset{(f)}{\leq} \sqrt{\frac{E_T}{P'_1}} \sum_{t=1}^{T} \|\boldsymbol{u}_{t+1} - \boldsymbol{u}_t\|$$

$$= \frac{1}{\sqrt{P'_1}} \sqrt{E_T} P'_T = \mathcal{O}(\sqrt{E_T}(P_T + 1)),$$

where $(a)$ is from $E_\tau \geq E_{\tau-1}$ for all $\tau$ by definition; $(b)$ holds similarly because $E_\tau$ is non-decreasing on $\tau$ (or $t$); $(c)$ holds because $P'_{\tau-1} \leq P'_\tau$ for all $\tau$ (for any slot $t$) and hence we can add additional positive terms and create the entire sum from $\tau = 2$ to $\tau = t$, instead of only the partial sum of terms in $[t]^+$; $(d)$ holds by writing the telescoping sum; $(e)$ holds by dropping the last term which is negative; and finally in $(f)$ we used the fact that $E_T \geq E_t, \forall t$.

### C.4. Proof of Theorem 3.4

Consider the following regularization strategy

$$\sigma = \frac{1}{8R^2}; \quad \sigma_t = \sigma \delta_t$$

$$\delta_1 = \langle \boldsymbol{g}_1, \boldsymbol{x}_1 \rangle - \min_{\boldsymbol{x} \in \mathcal{X}} \langle \boldsymbol{g}_1, \boldsymbol{x} \rangle; \quad \delta_t = h_{0:t-1}(\boldsymbol{x}_t) + \langle \boldsymbol{p}_t, \boldsymbol{x}_t \rangle - \min_{\boldsymbol{x} \in \mathcal{X}} \left( h_{0:t-1}(\boldsymbol{x}) + \langle \boldsymbol{p}_t, \boldsymbol{x} \rangle \right), \forall t \geq 2; \tag{33}$$

*Clarification on the term "recursive":* In previous regularization strategies, the strong convexity at time $t$, $\sigma_{1:t}$, can be expressed in closed form as $\sigma_{1:t} = \sigma \sqrt{E_t}$. This follows from defining each $\sigma_t$ to be exactly $\sigma(\sqrt{E_t} - \sqrt{E_{t-1}}) \leq \frac{\sigma \epsilon_t^2}{2\sqrt{E_{t-1}}}$. However, when $\sigma_t$ is defined more generally as a scalar $\sigma \delta_t$, where $\delta_t$ can take any value in $[0, \sigma(\frac{\epsilon_t^2}{2\sqrt{E_{t-1}}})]$, we lose this compact form. Instead, $\sigma_{1:t}$ is now recursively defined as $\sigma_{1:t-1} + \sigma \delta_t$. As discussed, this ensures minimal regularization, impacting both the algorithm's behavior and the analysis.

**Theorem 3.4.** *Under Settings 1, Alg. 1 run with the regularization strategy in* (33) *produces points* $\{\boldsymbol{x}_t\}_{t=1}^{T}$ *such that, for any $T$, the dynamic regret $\mathcal{R}_T$ satisfies:*

$$\mathcal{R}_T \leq 1.1\, \delta_{1:T} + \sum_{t=1}^{T-1} \frac{1}{4R} \delta_{1:t} \|\boldsymbol{u}_{t+1} - \boldsymbol{u}_t\| + H_T$$

$$\leq (3.7R + P_T)\sqrt{E_T} + H_T = \mathcal{O}\left( (1 + P_T)\sqrt{E_T} \right).$$

*Proof.* We begin from the result of Lemma 4.1 but without substituting the upper bound on part **(I)**, in order to characterize it more tightly via the $\delta_t$ terms. For the term **(II)**, we use the upper bound from (29):

$$\mathcal{R}_T \leq \sum_{t=1}^{T} (h_{0:t}(\boldsymbol{x}_t) - h_{0:t}(\boldsymbol{x}_{t+1}) + r_t(\boldsymbol{u}_t) - r_t(\boldsymbol{x}_t)) + \sum_{t=1}^{T-1} ((R\sigma_{1:t} + \|\boldsymbol{p}_{1:t}\|) \|\boldsymbol{u}_{t+1} - \boldsymbol{u}_t\|)$$

$$\overset{(a)}{\leq} \sum_{t=1}^{T} (h_{0:t}(\boldsymbol{x}_t) - r_t(\boldsymbol{x}_t) - h_{0:t-1}(\boldsymbol{x}_{t+1}) - r_t(\boldsymbol{x}_{t+1}) - \langle \boldsymbol{p}_t, \boldsymbol{x}_{t+1} \rangle + r_t(\boldsymbol{u}_t))$$

$$+ \sum_{t=1}^{T-1} \left( (2R\sigma_{1:t} + \epsilon_t) \| \boldsymbol{u}_{t+1} - \boldsymbol{u}_t \| \right)$$

$$\overset{(b)}{\leq} \sum_{t=1}^{T} \left( h_{0:t-1}(\boldsymbol{x}_t) + \langle \boldsymbol{p}_t, \boldsymbol{x}_t \rangle - h_{0:t-1}(\boldsymbol{x}_{t+1}) - \langle \boldsymbol{p}_t, \boldsymbol{x}_{t+1} \rangle + r_t(\boldsymbol{u}_t) \right)$$

$$+ \sum_{t=1}^{T-1} \left( (2R\sigma_{1:t} + \epsilon_t) \| \boldsymbol{u}_{t+1} - \boldsymbol{u}_t \| \right)$$

$$\overset{(c)}{\leq} \sum_{t=1}^{T} \left( \delta_t + r_t(\boldsymbol{u}_t) \right) + \sum_{t=1}^{T-1} \left( (2R\sigma_{1:t} + \epsilon_t) \| \boldsymbol{u}_{t+1} - \boldsymbol{u}_t \| \right)$$

$$\overset{(d)}{\leq} \delta_{1:T} + \frac{R^2 \sigma}{2} \delta_{1:T} + \sum_{t=1}^{T-1} 2R\sigma_{1:t} \| \boldsymbol{u}_{t+1} - \boldsymbol{u}_t \| + H_T$$

$$\overset{(e)}{\leq} 1.1\, \delta_{1:T} + \sum_{t=1}^{T-1} \frac{1}{4R} \delta_{1:t} \| \boldsymbol{u}_{t+1} - \boldsymbol{u}_t \| + H_T \tag{34}$$

where $(a)$ follows by Lemma 4.3 which bounds $\| \boldsymbol{p}_{1:t} \|$, and by the fact that $\sigma_{1:t-1} \leq \sigma_{1:t}$, $(b)$ by dropping the non-negative $-r_t(\boldsymbol{x}_{t+1})$, $(c)$ by the definition of $\delta_t$, $(d)$ by bounding each $\| \boldsymbol{u}_t \|$ in $r_t(\boldsymbol{u}_t), t \leq T$ by $R$, $(e)$ since $\sigma = 1/8R^2$.

### C.4.1. BOUNDING THE RECURSION

Note for each $\delta_t$ we have that

$$\delta_t \leq 2R\epsilon_t.$$

The above follows from Lemma B.2. Specifically, note that $\delta_t$ is equal to the expression in (23). In addition, we know from Lemma 4.2 that

$$\delta_t \leq \frac{\epsilon_t^2}{2\sigma_{1:t-1}},$$

by noticing that $\delta_t$ is the LHS in the inequality (25). Substituting the strong convexity term $\sigma_{1:t}$ for the choices made in this section in (33), we get

$$\delta_t \leq \frac{4R^2 \epsilon_t^2}{\delta_{1:t-1}}.$$

Overall,

$$\delta_t \leq \min \left( 2R\epsilon_t, \frac{4R^2 \epsilon_t^2}{\delta_{1:t-1}} \right).$$

We can now invoke the auxiliary Lemma D.4 on the last term with $a_t \doteq 2R\epsilon_t$, $\Delta_t \doteq \delta_{1:t}$ to get that for any $t$,

$$\delta_{1:t} \leq 2\sqrt{3} R \sqrt{E_t}.$$

Going back to (34), we get

$$\mathcal{R}_T \leq \frac{17\sqrt{3}}{8} R \sqrt{E_T} + \sum_{t=1}^{T-1} \frac{\sqrt{3}}{2} \sqrt{E_t} \| \boldsymbol{u}_{t+1} - \boldsymbol{u}_t \| + H_T$$

$$\leq \frac{17\sqrt{3}}{8} R \sqrt{E_T} + \frac{\sqrt{3}}{2} \sqrt{E_T} P_T + H_T = \mathcal{O} \left( (1 + P_T) \sqrt{E_T} \right).$$

$\square$

# D. Auxiliary Lemmas

Lemmas/theorems here are (specialized) results from the literature with potentially modified notation.

**Lemma D.1.** *(McMahan, 2017, Lemma 7) Given a convex function $\phi_1(\cdot)$ with a minimizer $y_1 \doteq \operatorname{argmin}_y \phi_1(y)$ and a function $\phi_2(\cdot) = \phi_1(\cdot) + \psi(\cdot)$ that is 1-strongly convex w.r.t some norm $\| \cdot \|$. Then,*

$$\phi_2(y_1) - \phi_2(y') \leq \frac{1}{2}\|b\|_*^2,$$

*for any $y'$, and any (sub)gradient $b \in \partial\psi(y_1)$.*

**Lemma D.2.** *(Auer et al., 2002, Lemma 3.5) For any $a_t \geq 0, \forall t \in [T]$*

$$\sum_{t=1}^{T} \frac{a_t}{\sqrt{a_{1:t}}} \leq 2\sqrt{\sum_{t=1}^{T} a_t},$$

*with the convention that $\frac{0}{\sqrt{0}} \doteq 0$.*

**Lemma D.3.** *(Orabona, 2022, Section 7.6) Let $R, \sigma$ be positive real numbers that satisfy $\sigma \leq \frac{1}{2b}$, $b > 0$. Then, for any $a_t \geq 0, \forall t \in [T]$:*

$$\sum_{t=1}^{T} \min\left(\frac{a_t^2}{2\sigma\sqrt{\sum_{\tau=1}^{t-1} a_\tau}}, ba_t\right) \leq \frac{\sqrt{2}}{\sigma}\sqrt{\sum_{t=1}^{T} a_t^2}.$$

*Proof.* The proof of this lemma closely follows the argument presented in the cited section. However, due to the inclusion of the term $b$ in the second part of the $\min$, we adapt the proof accordingly to account for this difference.

$$\sum_{t=1}^{T} \min\left(\frac{a_t^2}{2\sigma\sqrt{\sum_{\tau=1}^{t-1} a_\tau^2}}, ba_t\right) = \sum_{t=1}^{T} \sqrt{\min\left(\frac{a_t^4}{4\sigma^2 \sum_{\tau=1}^{t-1} a_\tau^2}, b^2 a_t^2\right)}$$

$$= \frac{1}{2}\sum_{t=1}^{T} \sqrt{\min\left(\frac{a_t^4}{\sigma^2 \sum_{\tau=1}^{t-1} a_\tau^2}, 4b^2 a_t^2\right)} \overset{(a)}{\leq} \frac{1}{2}\sum_{t=1}^{T} \sqrt{\frac{2}{\frac{\sigma^2 \sum_{\tau=1}^{t-1} a_\tau^2}{a_t^4} + \frac{1}{4b^2 a_t^2}}}$$

$$= \frac{1}{2}\sum_{t=1}^{T} \sqrt{\frac{2}{\frac{4b^2\sigma^2 \sum_{\tau=1}^{t-1} a_\tau^2 + a_t^2}{4b^2 a_t^4}}} \overset{(b)}{\leq} \frac{1}{2}\sum_{t=1}^{T} \frac{2\sqrt{2}\, b\, a_t^2}{\sqrt{4b^2\sigma^2 \sum_{\tau=1}^{t} a_\tau^2}} = \frac{\sqrt{2}}{2\sigma}\sum_{t=1}^{T} \frac{a_t^2}{\sqrt{\sum_{\tau=1}^{t} a_\tau^2}},$$

where $(a)$ follows by $\min(a, b) \leq \frac{2}{1/a + 1/b}$, and $(b)$ used the assumption on $\sigma$ to append $a_t^2$ to the sum. The result then follows by lemma D.2. $\qquad\square$

**Lemma D.4.** *(Orabona & Pál, 2018, Lemma 7) Let $a_t \geq 0, \forall t \in [T]$, and $\Delta_t$ be a sequence of positive numbers satisfying the recurrence*

$$\Delta_t = \Delta_{t-1} + \min\left(a_t, \frac{a_t^2}{\Delta_{t-1}}\right).$$

*with $\Delta_0 \doteq 0$. Then, for any $T$, we have that*

$$\Delta_T = \sqrt{3\sum_{t=1}^{T} a_t^2}.$$

# E. Comparison with the Literature

We summarize the key differences between our work and the relevant existing results on optimistic dynamic regret. The selected references represent the best known regret bounds (there are many other works that focus on different aspects, such as efficiency, unbounded domain, etc, but have the same structure as these bounds). It is important to note that some of these works derive bounds in terms of quantities other than the gradient prediction error, such as the extended "temporal variation" in (Scroccaro et al., 2023) or the "comparator loss" in (Zhao et al., 2024). For the sake of consistency, we restrict the comparison to bounds involving $E_T$, choosing $E_T$ in cases where a $\min(E_T, \cdot)$ term is present in the literature. Extending FTRL-based algorithms to handle comparator loss and temporal variations, dynamic regret bounds remain open. Lastly, in the comparison below, "best-case" refers to the scenario where $E_T = 0$, corresponding to perfect predictions.

**Bounds without tuning for $P_T$:**

- (Jadbabaie et al., 2015) Obtains a bound of $\mathcal{O}((P_T + 1)\sqrt{E_T + 1})$, which is $\mathcal{O}(P_T)$ in the best case.

- (Scroccaro et al., 2023) Obtains a bound of $\mathcal{O}((P_T + 1)\sqrt{D_T + 1})$, where $D_T \doteq \|\sum_{t=1}^{T} \nabla f_t(\boldsymbol{y}_{t-1}) - \nabla \tilde{f}_t(\boldsymbol{y}_{t-1})\|$. which is $\mathcal{O}(P_T)$ in the best case.

- (Zhao et al., 2024) Obtains a bound $\mathcal{O}(\sqrt{(V_T + P_T + 1)(1 + P_T)})$, which is $\mathcal{O}(P_T)$ in the best case. Note that all $P_T$ appear under the root. Hence, this bound still matches the optimal min-max bound of $\sqrt{T(1 + P_T)}$ in the worst case.

- This work achieves a dynamic regret bound of $\mathcal{R}_T = \mathcal{O}((1 + P_T)\sqrt{E_T}$, which is 0 in the best case.

**Bounds with online $P_T$ estimation**:

- (Jadbabaie et al., 2015) Obtains a bound of $\mathcal{O}(\log(T)\sqrt{(P_T + 1)(E_T + 1)})$, which is $\mathcal{O}(\log(T)\sqrt{P_T})$ in the best case.

- (Scroccaro et al., 2023) Obtains a bound of $\mathcal{O}(\sqrt{(P_T + 1)(D_T + \theta_T)})$, where $\theta_T$ is the sum of corrective terms $\theta_t$, which are added to the learning rate to ensure monotonicity, and it holds that in the perfect *gradient* prediction case $\theta_T = \mathcal{O}(1 + P_T)$ and the regret bound is also $\mathcal{O}(P_T)$.

- (Zhao et al., 2024) Maintains the same bounds since, as detailed in the main paper, this meta learning framework provides bounds that hold simultaneously for all $P_T$. The bound is $\mathcal{O}(\sqrt{(V_T + P_T + 1)(1 + P_T)})$, which is $\mathcal{O}(P_T)$ in the best case.

- This work achieves a dynamic regret bound of $\mathcal{O}((1 + \sqrt{P_T})\sqrt{E_T} + A_T)$, which is 0 in the best case.

# F. Numerical Examples

In this section, we present numerical examples comparing the performance of three standard implementations of OCO algorithms across multiple non-stationary environments (sequences of cost functions). The algorithms considered are:

- FTRL with adaptive Euclidean regularization, which corresponds to a lazy projected Online Gradient Descent (OGD) but with Adagrad style tuning (McMahan, 2017, Sec. 3.5)

- OMD with data-adaptive learning rates, which corresponds to a greedy projection OGD (Orabona, 2022, Sec. 4.2).

- Our proposed algorithm, `OptFPRL`, with the vanilla tuning strategy (i.e., in Sec. 3.1).

The implementation code for the algorithms, along with the code to reproduce all experiments, is available at the following repository: (Mhaisen, 2025). Since the FTRL and OMD variants used in our experiments neither assume prior knowledge of $P_T$ nor attempt to estimate it online, we compare them to `OptFPRL` using the tuning strategy described in Sec. 3.1 to ensure a fair evaluation. In scenarios where no predictions are used, the predicted functions fed to the algorithms are set to zero. This allows us to understand their performance initially, independent of prediction quality. Even without predictions, the proposed algorithm outperforms the two benchmarks in many scenarios, demonstrating its performance in dynamic environments.

**Numerical setup**. $\mathcal{X} \doteq \{\boldsymbol{x} \in \mathbb{R}^{16} | \|\boldsymbol{x}\| \leq 2\}$ $f_t(\boldsymbol{x}) = \langle \boldsymbol{c}_t, \boldsymbol{x} \rangle$, $T = 5000$, with

- Scenario 1: $\boldsymbol{c}_1, \ldots, \boldsymbol{c}_{1000} = -\mathbf{1}$, $\quad \boldsymbol{c}_{1000}, \ldots, \quad \boldsymbol{c}_{5000} = \mathbf{1}$.

- Scenario 2: $\boldsymbol{c}_1, \ldots, \boldsymbol{c}_{1000} = -\mathbf{1}$, $\quad \boldsymbol{c}_{2000}, \ldots, \boldsymbol{c}_{2500} = -\mathbf{1}$, $\quad \boldsymbol{c}_{3500}, \ldots, \quad \boldsymbol{c}_{3750} = -\mathbf{1}$, $\boldsymbol{c}_t = \mathbf{1}$ otherwise.

- Scenario 3: $\boldsymbol{c}_1, \ldots, \boldsymbol{c}_{1000} = -\mathbf{1}$, $\quad \boldsymbol{c}_{2000}, \ldots, \boldsymbol{c}_{2500} = -\mathbf{5}$, $\quad \boldsymbol{c}_{3500}, \ldots, \quad \boldsymbol{c}_{3750} = -\mathbf{10}$, $\boldsymbol{c}_t = \mathbf{1}$ otherwise.

- Scenario 4: $\boldsymbol{c}_t$ alternates between $\mathbf{1}$ and $\mathbf{-1}$ every $50$ steps.

- Scenario 5: $\boldsymbol{c}_t$ alternates between $\mathbf{1}$ and $\mathbf{-0.1}$ every $50$ steps.

- Scenario 6: $\boldsymbol{c}_t$ alternates between $\mathbf{1}$ and $\mathbf{-1}$ every $50$ steps; Predictions $\tilde{\boldsymbol{c}}_t = \boldsymbol{c}_t - \frac{\boldsymbol{c}_t}{0.1t}, \forall t \in [T]$

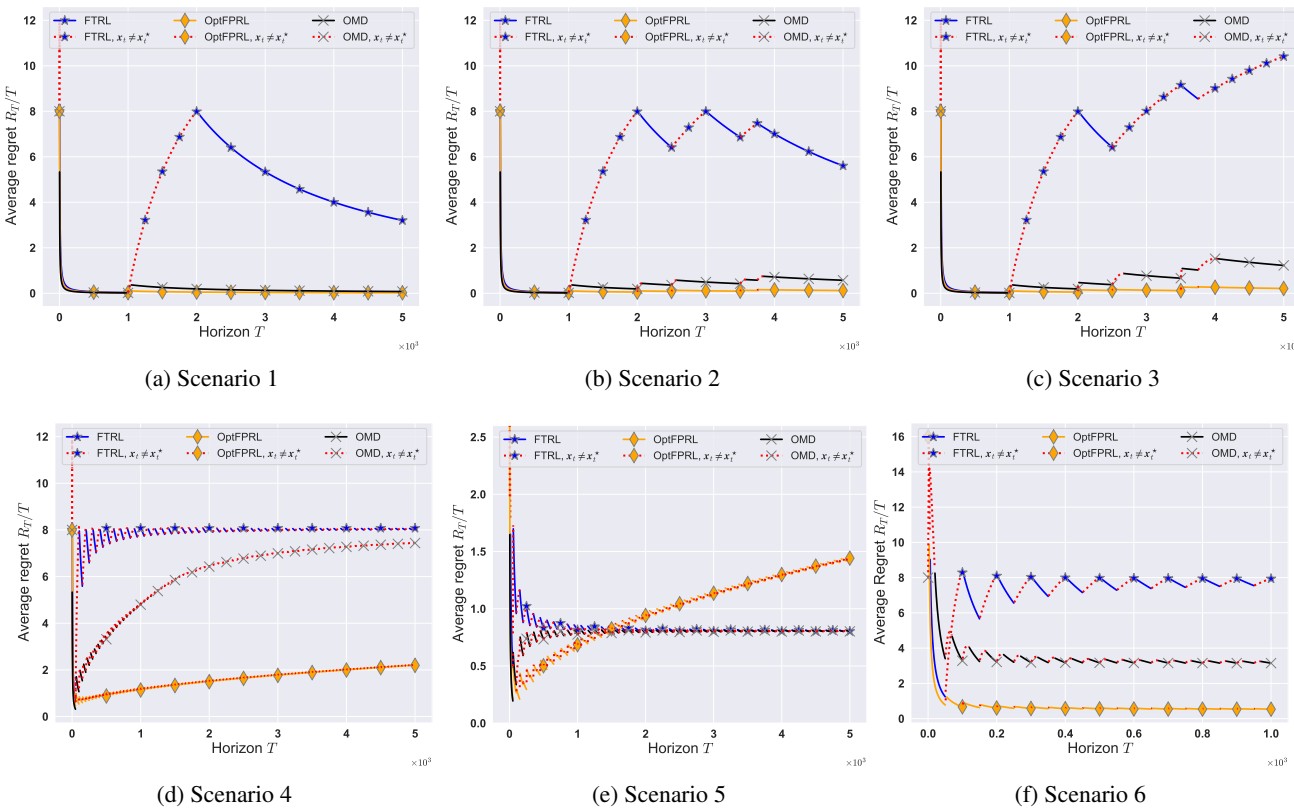

Figure 2: Average dynamic regret over time across various non-stationary scenarios. Dashed lines indicate time slots where the computed iterate differs from the comparator $\boldsymbol{x}_t^\star$.

The observed behavior across different scenarios in the above figure aligns with theoretical expectations. In Scenario 1, where the cost function shifts at $t = 1000$, standard FTRL struggles to adapt due to its reliance on accumulating all past costs. This inertia makes it slow to respond and leads to continued suboptimal actions until $t = 2000$, resulting in high regret. In contrast, OMD reacts immediately to the shift, adjusting its actions accordingly. Similarly, `OptFPRL` adapts directly, resulting in lower regret.

In Scenario 3, where cost directions change multiple times with increasing magnitudes, the limitations of FTRL become evident. The average regret fails to diminish, demonstrating its inability to handle such non-stationarity. While both OMD and `OptFPRL` respond to these variations, `OptFPRL` achieves lower regret. The advantages of `OptFPRL` are even more pronounced in Scenario 4, which involves high-frequency cost changes. That said, the observed difference between the implemented versions of OMD and `OptFPRL` is primarily due to the parameter tuning of each algorithm. The tested configurations use the theoretically optimal learning rate $\eta_t$ for OMD and regularization parameter $\sigma_t$ for `OptFPRL`, but different choices may lead to considerably different behavior. In contrast, the poor performance of vanilla FTRL cannot be mitigated by tuning theoretically motivated parameters–its failure is more fundamental, as discussed in the main text.

We also highlight a scenario in which `OptFPRL` performs the worst (Scenario 5): high-frequency cost switches with alternating magnitudes (large and small). This setting is deliberately designed to exploit our method's extra agility, forcing "undue" frequent adjustments. As expected, `OptFPRL` exhibits higher regret in this case, consistent with the theoretical results. Nonetheless, this tradeoff is inherent to the design and provides insights into potential extensions that balance agility and stability.

Lastly, we highlight in Scenario 6 the role of very high-quality predictions in the performance of the *optimistic* versions of the three algorithms ((Jadbabaie et al., 2015) for OMD and (Joulani et al., 2020, Sec. 7.1) for FTRL). We plot the average regret under predictions constructed as the original functions plus adversarial noise. Specifically, the adversarial noise is set as the negative of the original cost functions, with magnitude decaying quickly as $1/(0.1t)$, becoming negligible by $t \approx 100$. As noted in the paper, standard FTRL can be easily "trapped", accumulating redundant gradients and failing to track the comparators. Optimistic OGD and our OptFPRL react immediately when losses change direction.

