# OpenReview forum: "On the Dynamic Regret of Following the Regularized Leader: Optimism with History Pruning"
_ICML.cc/2025/Conference — ICML 2025 poster_

### Official Review · Reviewer_114P · 2025-03-05

**Overall Recommendation:** 3

**Summary:**

The paper presents a FTRL variant for dynamic regret with optimism in a compact domain. The paper relies on an adaptive correction $g_t^I$ to correct stored gradient states $\sum_t g_t$, similar as an ``adaptive restart'' of FTRL whenever is desired. This general development yields to regret bound when path length $P_T$ is known or unknown a priori, which are mimax optimal. The bound also automatically recovers optimal static data dependent regret.

**Claims And Evidence:**

Yes.
The paper claims investigating FTRL variants for dynamic regret, which is supported by their results in section 3.

**Essential References Not Discussed:**

Well referenced.

**Experimental Designs Or Analyses:**

Included in appendix F, most case works well support the hypothesis of the need of FTRL based dynamic regret. The author was also very open to discuss the worst case and give explanation.

**Methods And Evaluation Criteria:**

The paper use classical regret as evaluation criterion.

**Other Comments Or Suggestions:**

with potential typo in line 395, should be <=

**Other Strengths And Weaknesses:**

Strength: The paper is well written and easy to follow. The remarks under each theorem are very helpful to compare results to previous literature.

Weakness: It is in general questionable the need of FTRL based data dependent dynamic regret, as there are already other algorithms can achieve the same bound. Especially seeing the development in this paper are still fragile in scenario 5.

**Questions For Authors:**

None

**Relation To Broader Scientific Literature:**

As claimed, fill the gap of FTRL based data dependent dynamic regret.

**Theoretical Claims:**

main text was checked

---

> ### Author Rebuttal · Authors · 2025-04-01
>
> Thank you for taking the time to read the paper and providing the feedback
>
> **Weaknesses**.
> We appreciate the reviewer’s recognition of the clarity of the FTRL-based analysis for dynamic comparators. We understand the concern regarding its broader impact, given that the minimax dynamic regret bounds are known and achieved by other algorithms. Our motivation for introducing a new FTRL variant and analyzing it in this setting is as follows:
>
> $\bullet$ Flexible regularization. A key strength of FTRL is its ability to use general regularizers that can vary arbitrarily between iterations. For instance, proximal regularizers centered around different points in the domain allow the algorithm to incorporate prior knowledge about potentially good points, $\boldsymbol{\tilde{x}}_t$, as centers. This flexibility contrasts OMD, whose stability term is a scaled version of a fixed mirror map.
>
> $\bullet$ Flexible pruning:
> Our framework supports flexible pruning, enabling algorithmic variants between lazy and greedy methods. To see this point perhaps more clearly, the reviewer is kindly invited to take a look at the *Dual interpretation* asked by reviewer ecer.
> While the framework of Jacobsen and Cutkosky (2022) could, in principle, do that, we present here FTRL-native theoretical insights (Lemmas $4.1$\&$4.3$). These results provide explicit trade-offs and open avenues for adaptively choosing pruning frequency based on iterate-stability preferences.
>
> $\bullet$ Further insights on OMD/FTRL connections. The importance of further understanding the FTRL framework was recently highlighted by Ahn et al. (2024), who showed that the widely used Adam optimizer is essentially a form of FTRL with an exponentially decaying dual-space state. This motivates revisiting FTRL with potential modifications to how it retains this state. We see our work as a contribution in this direction, identifying when and why FTRL fails under dynamic comparators, and proposing selective pruning as a principled solution.
>
> Overall, we note that when one departs from the original versions of FTRL and OMD, it is not difficult to create variants of one algorithm that resemble (in formulas or bounds) the other. Independent of naming and definitions, our work shows that pruning the history of gradients enables dynamic regret bounds with desirable properties. It also opens new directions for exploring interpolation between laziness and greediness, which lies at the core of this very interesting FTRL vs OMD discussion.
>
> **Other comments**.
>
> $\bullet$ 395: it is indeed an inequality and not equality.
> Thanks for pointing out the typo.

---

### Official Review · Reviewer_oZgn · 2025-03-12

**Overall Recommendation:** 4

**Summary:**

This paper investigates the Follow-the-Regularized-Leader method (FTRL) under the Online Convex Optimization (OCO) framework for dynamic regret minimization. Specifically, the authors propose a series of FTRL-based methods, deliver novel analysis and establish new dynamic regret bound, which can recover existing known results. Overall, the topic is crucial for OCO community, and the theoretical guarantees are interesting.

**Claims And Evidence:**

Yes, there are experiments to verify the proposed theoretical results.

**Essential References Not Discussed:**

This work primarily investigates dynamic regret optimization, a highly active topic in the online learning community. I suggest that the authors include the following studies in the related work section to better illustrate the development of dynamic regret optimization.

* Optimal Dynamic Regret in Exp-Concave Online Learning. 2021.
* Optimal Dynamic Regret in Proper Online Learning with Strongly Convex Losses and Beyond. 2022.
* Non-stationary Projection-Free Online Learning with Dynamic and Adaptive Regret Guarantees. 2024.

Additionally, I noticed that the authors mention the SEA model but only discuss the work of Chen et al. (2024), overlooking other important milestones. I suggest that the authors incorporate discussions of the following works in their introduction about the SEA model.

* Between stochastic and adversarial online convex optimization: Improved regret bounds via smoothness. 2022.
* Online Composite Optimization Between Stochastic and Adversarial Environments. 2024.

Finally, since the topic of this paper is highly significant, I suggest that the authors use a table to systematically compare previous works with the contributions of this paper.

**Experimental Designs Or Analyses:**

There are experimental studies in the appendix to support theoretical findings.

**Methods And Evaluation Criteria:**

Yes.

**Other Comments Or Suggestions:**

* In Line 084, there appears to be a citation error. In Chen et al. (2024), Remark 11 does not seem to discuss dynamic regret optimization for FTRL-based methods.
* In Line 181, typos: JC22 => Jacobsen & Cutkosky, 2022
* In Line 265, typos: $E'_T=O(T^)$ => $E'_T=O(T)$

Below are some formatting suggestions:

* I recommend avoiding the multiple use of footnotes.
* The authors place the experimental section in the appendix. I suggest considering its inclusion in the main part instead.

**Other Strengths And Weaknesses:**

**Strengths**

* This paper is well-organized, clearly written, and easy to follow.
* This paper presents an insightful understanding for the literature. The authors provide a comprehensive explanation on the motivation behind the proposed method.

**Weakness**

* See in the following parts.

**Questions For Authors:**

* In Line 302, what is the meaning of $1+ \beta - 1\leq 1/2+\beta/2$？In addition, I suggest explicitly stating that $P_T=O(T^{\beta})$ at this point for clarity.
* In my view, the construction of $g_t^I $ in (5) is crucial for the analysis. The authors could further elaborate on the motivation behind this design—whether it arises from analytical requirements or is inspired by existing works.

**Relation To Broader Scientific Literature:**

This work bridges the gap between dynamic regret minimization and FTRL-based methods, a topic that has been explored in several related studies (Jacobsen & Cutkosky, 2022).

**Theoretical Claims:**

I review the overall proof framework, and its analysis follows some existing analytical tools.

---

> ### Author Rebuttal · Authors · 2025-04-01
>
> Many thanks for your review and feedback.
>
> **References**.
> We selected Chen et al. (2024) from the SEA thread as it was the first to study SEA under the dynamic regret metric. That said, we will update the related work with the suggested background references (SEA \& others) so as to better reflect the development of dynamic regret, and include a tabular summary of bounds.
>
> **Other comments**.
>
> $\bullet$ Line 84: You are right, the discussion appears after Remark (8);  Remark (11) was from their conference version. We will correct this in the revision.
>
> **Questions**.
>
> $\bullet$ Regarding the inequality, this should be just $\beta\leq1/2+\beta/2$ to highlight that the $H_T \doteq \sum_t \epsilon_t \lVert \boldsymbol u_t - \boldsymbol u_{t+1} \rVert$ term does not dominate the $\sqrt{P_TE_T}$ term in worst case prediction error $E_T=c E'_T= O(T)$. Note that in light of the discussion with reviewer 2ZjR (in the Remark), we will update the remark by mentioning that $H_T$ is $\mathcal{O}\sqrt{P_TE_T}$
>
> **Design choice**.
> Below, we outline the rationale behind our pruning design from three complementary perspectives: preliminary intuition, analytical basis, and suggestive related work.
>
> $\bullet$  *Intuition*. The pruning rule is motivated by the primal-dual view in online learning over compact sets. While primal iterates stay within the compact domain $\mathcal{X}$, dual variables (e.g., cumulative gradients) can grow unbounded in $\mathbb{R}^n$. If projections repeatedly return the iterate to the same region, why keep accumulating dual information that no longer influences decisions? Pruning addresses this by trimming dual history when appropriate. The pruning cone adds flexibility, enabling partial or delayed resets to control stale information in the dual space. Of course, choosing to prune at each time makes the update equivalent to OMD (see related work below).
>
> $\bullet$ *Analytical basis*. If we wish to manipulate the update $\boldsymbol{x}_t$ to offset accumulated gradients, the natural starting point is the linearization of $f_t(\cdot)$, as it directly shapes the update. Ideally, we seek predictions $\tilde f_t(\cdot)$ whose subgradients can cancel or reduce the influence of past gradients. This is precisely what the pruning cone captures: it characterizes the directions in which the update can afford to remove history, and the composition with the indicator function formalizes this capability.
>
> $\bullet$ *Related work*.
> As discussed in the paper, a key inspiration comes from the equivalence between OMD and FTRL, first articulated in McMahan’s survey, for unbounded domains, and for bounded domains under fixed learning rates. This equivalence was further explored by Fang et al. (2022) [Online Mirror Descent and Dual Averaging…], who extended the analysis to time-adaptive learning rates. Both works focus on non-optimistic static regret. Additionally, the comprehensive work of Jacobsen \& Cutkosky (2022) introduced the centered mirror descent family, capturing both lazy and greedy updates by composing the indicator with either their $\psi_t$ or $\varphi_t$ terms. This naturally raises the question of how varying this choice across time may influence the algorithm’s behavior, and what roles predictions play in this.

---

### Official Review · Reviewer_ecer · 2025-03-13

**Overall Recommendation:** 3

**Summary:**

This work revisits the Follow-The-Regularized-Leader (FTRL) framework and explores how to utilize FTRL to derive dynamic regret bounds. The key finding in this paper is to predict with the modified loss $f_t(x) + I_{X}(x)$ rather than $f_t(x)$, where $I_X(x)$ is the indicator function. When the unprojected decision $x_t^{uc}$ in the FTRL framework deviates from the domain $X$, the introduction of the indicator function $I_X$ serves as a  "corrective mechanism", thus enabling the derivation of dynamic regret bounds.

This paper considers two cases: when the path length $P_T$ is known and when $P_T$ is unknown in previous but the comparator $u_t$ at each round can be perceived. For the former case, an $O((1+P_T)\sqrt{E_T})$ bound is established, where $E_T$ is a data-dependent factor. As for the latter case, the authors achieve an $O((1+\sqrt{P_T})\sqrt{E_T} + A + \min\{P_T + E_T^\prime\})$ bound, where $A$ could be  $O((1+P_T)\sqrt{E_T})$ at the worst case.

**Claims And Evidence:**

In Line 105, the authors claim "As for FTRL, to the best of our knowledge, no dynamic regret guarantees, whether problem-dependent or not, have been established in the existing literature". While in the paper  "Understanding Adam Optimizer via Online Learning of Updates: Adam is FTRL in Disguise" published in ICML’24, this paper shows that dynamic bounds can be obtained through the FTRL framework by adding discounted factors to the losses. Conducting a more comprehensive survey and provide a more detailed comparison with the metioned approach is necessary.

**Essential References Not Discussed:**

An essential reference missing from this paper is [Understanding Adam Optimizer via Online Learning of Updates: Adam is FTRL in Disguise, ICML’24], which is the pioneering work to establish dynamic regret bounds under the FTRL framework.

**Experimental Designs Or Analyses:**

I checked the experimental designs, which is acceptable to support their findings.

**Methods And Evaluation Criteria:**

The authors present a toy numerical experiment to support their findings, which is acceptable given that this paper primarily emphasizes the theoretical understanding of the FTRL framework.

**Other Comments Or Suggestions:**

- Is $\partial$ missed in Line 100 when recalling $g_t$?
- Is the definition of V_T on line 109 missing a squared term?
- There is a typo ($O(T)$) at the right column in Line 266.
- I am confused about the definition of $A$ at the right column in Line 276. Are there any typos?

**Other Strengths And Weaknesses:**

Strengths:

- The structure of this paper is well-organized and it is easy-to-follow.
- This paper explores an interesting direction, as the relationship and differences between FTRL and OMD is crucial to realize two fundamental online learning algorithmic designs.

Weakness:

- The assumption of knowning $u_t$ at each round is too strong at least for me, and optimizing the exact values of $u_t$ may lead to overfitting to the online loss functions.
- Theorem 3.3 is not optimal at the worst case, making it less competitive to existing results.
- The obtained results are not guaranteed to hold for any comparators.

Overall, this work presents interesting views to the FTRL framework. However, I believe it could be further refined by considering more general cases of dynamic regret bounds, which would help enhance its impact.

**Questions For Authors:**

- Can the introduction of indicator functions be understood from a primal-dual perspective? From this viewpoint, OMD performs a non-projected update in the dual space and only applies projection when transitioning back to the primal space. Could a similar perspective be applied to FTRL?
- Is it possible to modify the proposed algorithm so that it can guarantee dynamic regret bounds for arbitrary comparators?

**Relation To Broader Scientific Literature:**

The similar ideas of forgetting or pruning some information of the FTRL algorithm to derive dynamic regret bounds have been proposed by [Understanding Adam Optimizer via Online Learning of Updates: Adam is FTRL in Disguise, ICML’24] through adding discounted factors to the losses. This paper considers an another approach by adding indicator functions to the loss functions.

**Theoretical Claims:**

I checked the key proof highlighted in the main text.

---

> ### Author Rebuttal · Authors · 2025-04-01
>
> Thank you for reading the paper and providing feedback.
>
> **Claims \& missing reference**.
> We agree that it is important to discuss carefully Ahn et al. “Adam is FTRL in …”; we indeed cite this paper and mention that it uses an FTRL variant that attenuates history to obtain discounted regret bounds. That said, their paper also introduces a reduction from discounted to dynamic regret, which we did not discuss in detail in the current draft. Thanks for pointing this out. Here is a brief clarification:
>
> $\bullet$ In the bounded domain setting, their bound is $P_T^{1/3} T^{2/3}$ (From their Corollary 3.4, assuming knowledge of $P_T$), which is suboptimal due to its super-$\sqrt{T}$ dependence.
>
> $\bullet$ In contrast, our bounds, which also assumes $P_T$ knowledge, achieve $\sqrt{P_T T}$ in the worst case and further establish a refined  $\sqrt{P_T E_T}$, which is data-dependent.
>
> This is not a limitation of their work, but rather a natural outcome of their goal: explaining ADAM via FTRL.
>
> To clarify this, we will:
> $(i)$ explicitly attribute the first *sublinear-in-T* dynamic regret bound for FTRL to Ahn et al. (2024), and
> $(ii)$ revise the sentence in our paper to read:
> “…As for FTRL, no dynamic regret guarantees, data-dependent or otherwise, *with $\sqrt{T}$ rate* have been established ...”
>
> We note that their Cor. 3.5 improves the bound to $\sqrt{P_T T}$ but only in the unbounded setting.
>
> **Weakness 1\&3**: Note that Theorems $3.1$,$3.2$, and $3.4$, do in fact hold for all comparators (i.e., without knowing $\boldsymbol{u}_t$). This is because in Thm. $3.1$ and $3.4$, no information on $P_T$ is used (and the price is paid accordingly). As for Thm. $3.2$, knowing $P_T$ in advance *does not* necessarily mean that we know all the comparators, but rather that we compete with sequences whose path length is at most $P_T$. Here, we followed the standard notation from the literature, but indeed, a better notation would be fixing any $B$. Then, we compete with *any* sequence whose $P_T\leq B$ to get $\sqrt{BE_T}$.
>
> It is only Thm. $3.3$ that assumes observability of $\boldsymbol u_t$ online. Yet, these need not be the minimizers of each corresponding $f_t$ (i.e., not necessarily overfitting).
>
> **Weakness 2**:
> For observable comparators, estimating $\sqrt{E_T / P_T}$ online is infeasible due to non-monotonicity. Our data-dependent approach recovers the $\sqrt{E_T P_T}$ bound when $\sqrt{P_t/E_t}$ changes direction $O(\sqrt{P_T})$ times (i.e., mostly monotone). The method performs no worse than tuning strategy $1$, while offering improvements when this structure is present. In all cases, the doubling trick of Jadbabaie et al. remains applicable.
>
> **Other comments**
> Thanks for spotting the typos. For the def. of $A$, it is more convenient to keep it but update the proof accordingly to have $2R\sigma A$ instead of $A/2$.
>
> **Questions**.
>
> **Dual perspective**. To provide a dual view, we look at the update step via the lens of dual maps. Recall the standard FTRL update:
> $
> \boldsymbol x_{t+1}^{\text {RL}} = \arg\min_x \langle \boldsymbol g_{1:t}, \boldsymbol x\rangle + r_{0:t}(\boldsymbol x)
> =\nabla r_{0:t}^*(-\boldsymbol g_{1:t})$,  where $r_{0:t}^*(\cdot)$ is the conjugate of the dual map $\nabla r_{1:t}(\cdot)$, restricted to $\mathcal{X}$ via $r_0 = I_\mathcal{X}$. From this viewpoint, FTRL maintains the state as cumulative gradients in dual space.
>
> In contrast, the OMD update, as you noted, is expressed as: $\boldsymbol x_{t+1}^{\text{MD}} = \nabla r_{0:t}^*(\nabla r_{1:t-1}(\boldsymbol x_t) - \boldsymbol g_t)$.
>
> Our selective pruning update can be simply interpreted as:
> $\boldsymbol x_{t+1} = \arg\min_x \langle  \boldsymbol p_{1:t}, \boldsymbol x \rangle + r_{0:t}(\boldsymbol x) = \nabla r_{0:t}^*(-\boldsymbol p_{1:t}) = \nabla r_{0:t}^*(\nabla r_{1:t-k-1}(\boldsymbol x_{t-k})-\boldsymbol g_{t-k:t})$,
> where $k$ denotes the most recent step at which we chose to prune. The last equality holds by the definition of $\boldsymbol g_k^{I}$ (assuming no predictions).
> Intuitively, we retain explicit gradient history only since the last pruning step $t-k$, while summarizing earlier history implicitly via the dual mapping of $\boldsymbol{x}_{t-k}$. The crux of the paper is showing that the way history is split (explicitly tracked after pruning, and implicitly captured before) is what controls dynamic regret.
>
> **Guaranteeing** $\sqrt{P_TE_T}, \forall u_t$. One can ``modify" OptFPRL with the second tuning strategy (that competes with comparators of at most $P_T$ switches); we can instantiate a set of $\Theta(\log T)$ sub-learners, each with a halving $\sigma$ starting from $1/\sqrt{T}$. Then, using the meta-learner of Zhao et al. (2020), the minimax bound can be recovered. The gist of this approach is that eventually $\exists$ an expert $i$ such that $\forall P_T, \sigma^{(i)} \geq 1/\sqrt{P_T}\geq 1/2\sigma^{(i)}$. This will be added to the paper as part of the discussion.

---

> > ### Comment · Reviewer_ecer · 2025-04-02
> >
> > I thank the authors for their responses, which address my concerns. Accordingly, I have decided to raise my score to 3. However, I believe that, to enhance the impact of the results, the paper should provide a more thorough discussion on how to use the meta-learner proposed by Zhao et al. (2020) to achieve the $\sqrt{P_T E_T}$ result. I hope the authors will revise the manuscript accordingly.

---

> > > ### Author Response · Authors · 2025-04-02
> > >
> > > Thank you for taking the time to engage with our response and for updating the score. We appreciate your feedback and will ensure that the points raised in our exchange are incorporated into the final version to further strengthen the paper.

---

### Official Review · Reviewer_2ZjR · 2025-03-14

**Overall Recommendation:** 4

**Summary:**

The paper presents a new optimistic algorithm, Follow-The-Pruned-Leader (FPRL) that aims to achieve dynamic regret in $O(\sqrt{P_TE_T})$, where $E_T$ measures the prediction error. The key insight is to avoid simply stacking previous gradients as they make the standard Follow-the-Regularized-Leader (FTRL) less adaptive to dynamic environments. It is done by "pruning" past gradients when their updates become too "large". They achieve $O(\sqrt{P_T E_T})$ when $P_T$ is known in advance or the comparison  sequence is observable, but also universal regret (i.e for all sequences simultaneously). Finally, all the terms in the regret scale with the prediction error, meaning that, compared to previous work who had an incompressible dependence on $P_T$, the dynamic regret of FPRL can be constant for perfect predictions.

## update after rebuttal

I decided to maintain my score

**Claims And Evidence:**

Yes, They provide proofs for all their claims along with simulation experiments the improvement of their method.

However, the claims regarding the remark 258R-270R is not detailed in appendix contrary to the authors claims.

**Essential References Not Discussed:**

None that I can think of.

**Experimental Designs Or Analyses:**

Yes, the experimental design makes sense and highlight the strength and weaknesses of their method. It would have been good to also have experiments showing how the regret converges for perfect predictions while other methods that have a linear dependency on $P_T$ keep increasing. This would be in a very dynamic environment, like scenario 4 of Appendix F

**Methods And Evaluation Criteria:**

They use dynamic regret as a criteria which is standard.

**Other Comments Or Suggestions:**

A bit too many typos and math alignments that are wrong: (I use L for the left column and R for the right)
1. the notation $z_{1:t} := \sum_{\tau=1}^t z_\tau $ where $z$ is any sequence is never formally introduced. (yes it appears in line 72R but not as a definition).
2. The function $h$ is used in line 307R-310R before being introduced in line 367L).
3. Appendix 180: I think it should be $x_t  = \arg\min_{x\in\mathcal{X_t}}$ instead of for any $x_t \in \mathcal{X}, \dots$. As the condition (5) in line 185 is true only if $x_t$ is the minimizer of $\mathcal{X}$, but I might be wrong
4.  Math Alignment issues ( the equal or inequality signs appear mid lines, making it hard to read):
- Appendix 324-326
- 385R-389R
- Appendix 165 . That $\tilde f_{T+1}$ term is too isolated
5. Typos
- 265R ($E(T^)$ (missing the power and the parenthesis is raised
- Appendix 638 $\delta_t$ appears in its own upper bound
- Appendix 660 $<p_t, x> \to <p_t, x_t>$
- 421L . "we have $R_T$" followed by equations. Maybe it should be "$R_T$ is upper bounded by:" or $R_T$ should be in the equation below.

**Other Strengths And Weaknesses:**

Strengths:
- The experiments really highlight the benefit of their method
- The regret derivation is easy to follow and the main body explain the core ideas pretty well


Weaknesses:
- Some math equations are hard to follow because of typos and alignments.
- Computation cost (see next weakness)
- The authors mention avoiding the circular dependency, but I am unsure (if we want to stay computationally efficent). OOMD linearize first in order to minimize computation cost, and since $x_t$ is unknown, the linearization is often done around $\tilde x_t$ (an intermediate value that does not depend on $f_t$). Here $x_t$ depends on $\tilde f_t$, thus we cannot linearize around $x_t$ to make the computation. So, if we wanted to stay computationally efficient and linearize the prediction, it will be around $x_{t-1}$ leading to the same issue noted by Scroccaro et al. (2023).

**Questions For Authors:**

1. What is the difference in computation time between this method and OMD. Does the method benefits from using an optimization on $\tilde f_{t+1}$ when OOMD focuses on the linear approximations.
2. (Repeat on the experiment) Does the regret converges for perfect predictions while other methods that have a linear dependency on $P_T$ keep increasing. (The regret, not the average regret).
3. See third weakness. Any comment ?
4. In appendix 103, you say, when $\sigma_{1:t} = 0$ but it does not seem to be used in that proof. Is that a typo?

**Relation To Broader Scientific Literature:**

- The algorithm builds on previous FTRL / OptFTRL work but contribute a new idea of choose smartly a subgradient from $I_\mathcal{X}$ (indicator function of the set) to have a more adaptive algorithm. Their update has some similarity with (McMahan, 2017)
- Their dynamic  regret guarantee improves the current best ones, i.e Zhao et al., 2024 by having a regret that goes to a constant when the predictions are perfect.

**Theoretical Claims:**

I checked all proofs, all except the proof of Theorem 3.3 that I did not read. I skimmed through the one of Theorem 3.4.
No issues on what I read in terms of proof, but there is some comments that must be addressed, notably on the cyclic dependency (See weaknesses).

---

> ### Author Rebuttal · Authors · 2025-04-01
>
> Thanks for taking the time to read the paper and provide the feedback.
>
> **Claims \& evidence**
>
> $258R$: Indeed, these details were omitted from the main text due to space constraints and accidentally left out of the appendix. We apologize for this oversight. Briefly, the claim is that the term $H_T \doteq \sum_t \epsilon_t \lVert\boldsymbol u_{t+1} - \boldsymbol u_t\rVert $ is $\mathcal{O}(\sqrt{P_TE_T})$, and therefore never dominates  the bounds in Theorems $1$-$4$, since those are $\Omega(\sqrt{P_TE_T})$. We state the remark below.
>
> Remark: Let $\epsilon_t^2$ scale as ${1}/{t^{1-\alpha}}, \alpha \in (0,1]$ so that $E_T = \Theta(T^{\alpha})$. Similarly, let $\lVert \boldsymbol{u}_{t+1} - \boldsymbol u_t  \rVert$ scale as  $\frac{1}{t^{1-\beta}}, \beta \in (0,1]$ so that $P_T = \Theta(T^{\beta})$. Then, we get that $H_T  = \sum_t {1}/{t^{(1-\alpha)/2\ +1-\beta}} = \Theta(T^{\beta +\ \alpha/2\ -1/2})$ and it holds $
> {\beta +\ \alpha/2\ -1/2}\ \leq\ (\alpha+\beta)/{2},\quad \forall \alpha, \beta \in [0,1).$
> The RHS is the growth rate of $\sqrt{P_TE_T}$.
>
>
> **Weaknesses**.
>
> **On the circular dependency**. Incorporating non-linearized  $\tilde f_t(\cdot)$ is offered as an option to avoid circular dependencies, but it is not required for the update to be well-defined.  When efficiency is a priority, linearized FTRL, like OOMD, can follow the standard practice by setting  $\tilde {\boldsymbol g_t} = \nabla \tilde f_t(\boldsymbol x_{t-1}) $ (i.e., linearizing the predicted function around the previous iterate), as done in prior work (e.g., see $\tilde{\boldsymbol g_t}$  Flaspohler et al., 2021). Hence, we maintain linearized forms and computational efficiency at the cost of having the somewhat less-informed quantity $\sum_t \lVert \nabla f_t(\boldsymbol x_t) - \nabla \tilde f_t(\boldsymbol x_{t-1})\rVert^2$ instead of $E_T$, as you correctly note. Moreover, in many cases, avoiding non-linearization does not compromise efficiency (see Questions)
>
>
> **Questions**.
>
> **Linearization \& Efficiency**. In fact, including  $\tilde f_t(\cdot)$ does not always compromise efficiency. For example, when $\tilde f_t(\cdot)$ are weighted $\ell_2$-norms, the complexity remains the same as OMD (i.e., closed form in $\mathbb R^n$ + Bregman projection). Nonetheless, for general convex predictions, solving a convex subproblem is indeed required. In such cases, practitioners retain the option to linearize (around the previous iterate).
>
> **on** $\sigma_{1:t}=0$. This condition addresses the situation when $\boldsymbol x^{\text{uc}}_t$ does not exist. In other words, we simply specify that if such a point does not exist, it is treated as lying outside $\mathcal{X}$, ensuring the update step formula remains well-defined.
>
> **Recommended experiment**.
> We repeat Scenario 4 from the paper, but now comparing the optimistic variants of OGD, FTRL, and our proposed OptFPRL. We plot the regret $R_T$ under predictions constructed as the original functions plus adversarial noise. Specifically, the adversarial noise is set as the negative of the original cost functions, with magnitude decaying quickly as $1/(0.1t)$, becoming negligible by $t \approx 100$.
>
> The resulting figure, along with algorithm implementations and the full code to reproduce these results, is available at the doubly-anonymized link: https://anonymous.4open.science/r/11679-3538.
>
> As noted in the paper, standard FTRL can be easily “trapped,” accumulating redundant gradients and failing to track the comparators. Optimistic OGD and our OptFPRL react immediately when losses change direction. Indeed, achieving constant regret under perfect predictions primarily serves as a sanity check for scale-freeness, rather than the central advantage. This objective was set to preserve the scale-freeness property observed in OMD while developing FPRL. The main benefit of FTRL-based formulation lies in its flexibility to arbitrarily choose and, potentially center differently, regularizers, enabling a ``compressible" dependence on $P_T$, as opposed to having fixed mirror maps that time varying weights (i.e., learning rates).
>
> **Other comments**.
> Thank you for the careful reading, we will make sure to fix the typos.
>
> $\bullet$ Appendix $180$. True, the minimizer in $\mathcal{X}$ ($h_0=I_{\mathcal{X}}$).
>
> $\bullet$ Appendix $638$. $\delta_t \rightarrow \epsilon_t$.
>
> $\bullet$ $421$L. $R_T \rightarrow R_T\leq$.

---

### Decision · Program_Chairs · 2025-05-01

**Decision:**

Accept (poster)

**Comment:**

This paper presents techniques for employing the FTRL algorithm family to obtain dynamic regret, which is usually only obtained via mirror descent style algorithms. While the resulting regret bounds are not in of themselves new, the techniques may shed new light on the fundamental techniques used in deriving these bounds, and so will be of interest to the ICML community.